# Celiac Disease and Liver Damage: The Gut–Liver Axis Strikes Back (Again)? A Retrospective Analysis in the Light of a Literature Review

**DOI:** 10.3390/nu17010085

**Published:** 2024-12-28

**Authors:** Aurelio Seidita, Federica Latteri, Mirco Pistone, Alessandra Giuliano, Luca Bertoncello, Giorgia Cavallo, Marta Chiavetta, Francesco Faraci, Alessia Nigro, Alessandro Termini, Laura Verona, Agnese Ammannato, Salvatore Accomando, Francesca Cavataio, Maria Letizia Lospalluti, Michele Citrano, Diana Di Liberto, Maurizio Soresi, Pasquale Mansueto, Antonio Carroccio

**Affiliations:** 1Internal Medicine Unit, “V. Cervello” Hospital, Ospedali Riuniti “Villa Sofia-Cervello”, Via Trabucco, 180, 90146 Palermo, Italy; aurelio.seidita@unipa.it (A.S.);; 2Department of Health Promotion Sciences, Maternal and Infant Care, Internal Medicine and Medical Specialties (PROMISE), University of Palermo, Piazza delle Cliniche, 2, 90127 Palermo, Italy; 3Institute for Biomedical Research and Innovation (IRIB), National Research Council (CNR), 90146 Palermo, Italy; 4Gastroenterology Unit, “V. Cervello” Hospital, Ospedali Riuniti “Villa Sofia-Cervello”, Via Trabucco, 180, 90146 Palermo, Italy; 5Department of Pediatrics, University Hospital of Palermo, 90134 Palermo, Italy; 6Pediatric Gastroenterology Unit, “Di Cristina” Hospital, Palermo, 90134 Palermo, Italy; 7Pediatrics Unit, “V. Cervello” Hospital, Ospedali Riuniti “Villa Sofia-Cervello”, Via Trabucco, 180, 90146 Palermo, Italy; 8Department of Biomedicine, Neurosciences and Advanced Diagnostics (BIND), Institute of Biochemistry, University of Palermo, 90127 Palermo, Italy

**Keywords:** gluten, celiac disease, NAFLD, MAFLD, MASLD, hepatic steatosis, metabolic syndrome

## Abstract

**Background/Objectives**: An increasing number of studies have reported liver involvement in both children and adults with celiac disease (CD). This often manifests as isolated hypertransaminasemia or hepatic steatosis (HS). The aim of this study was to define the prevalence of hypertransaminasemia and HS in a pediatric population with CD before starting a gluten-free diet (GFD) and to analyze how the introduction of a GFD could modify this condition. We also conducted a state-of-the-art literature review of the association between hypertransaminasemia, metabolic dysfunction-associated steatotic liver disease (MASLD) and CD. **Methods**: We retrospectively reviewed the clinical charts of pediatric CD patients diagnosed in three different pediatric units of Sicily, analyzing clinical, laboratory, ultrasound, and histology data before and 12 months after the introduction of a GFD. **Results**: A total of 160 patients (65.0% females, median age 6.4 (0.8–13.2) years) were included; hypertransaminasemia and HS prevalences at diagnosis were 8.1% and 6.1%, respectively. Subjects with hypertransaminasemia were younger (*p* = 0.01) than those without and had higher frequencies of HS (*p* = 0.034) and anti-tissue transglutaminase (tTg) immunoglobulin (Ig)G positivity (*p* = 0.046). Subjects with HS were younger (*p* = 0.0001) and had a higher frequency of hypertransaminasemia (*p* = 0.029) compared to non-steatotic ones. After 12 months of a GFD, hypertransaminasemia and HS persisted in 53.8% and 50.0% of patients, respectively. **Conclusions**: The prevalences of hypertransaminasemia and HS in Sicilian pediatric CD patients seem to be lower than those reported in other geographical areas. A GFD can reverse the trend of liver involvement, although periods of longer than 12 months may be necessary. However, a GFD has been associated with an increased prevalence of HS, and so regular follow-up involving a nutritionist should be recommended to guide physicians in patient management.

## 1. Introduction

The definition of non-alcoholic fatty liver disease (NAFLD) presents a number of limitations. In fact, it has been proven that in both children and adults, NAFLD is strictly associated with a metabolic syndrome (MetS), and so the term metabolic-associated fatty liver disease (MAFLD) was first proposed instead [1,2] to define a condition of hepatic steatosis (HS) in addition to the presence of overweight or obesity, diabetes mellitus, or metabolic dysfunction [3,4]. More recently, it was renamed metabolic dysfunction-associated steatotic liver disease (MASLD) [5], a steatotic liver disease (SLD) in the presence of one or more cardiometabolic risk factor(s) and the absence of harmful alcohol intake [6].

The current clinical practice guidelines (CPGs) for the diagnosis, treatment, and follow-up of individuals with MASLD have been developed as a joint effort by the European Association for the Study of the Liver (EASL), European Association for the Study of Diabetes (EASD), and European Association for the Study of Obesity (EASO) [5], updating the multi-society NAFLD CPGs released in 2016 [7]. MASLD has become the most common chronic liver disease, and its prevalence will likely continue to rise. The estimated global prevalence of MASLD in the general population has risen from 25% in 2016 [8] to currently more than 30%, and its incidence is continually increasing [8,9,10]. It has been estimated that in approximately 10–30% of subjects with isolated steatosis progress to steatohepatitis and advanced liver disease, but this risk is much higher in the presence of type 2 diabetes (T2D) (42–65% have steatosis). The MASLD spectrum comprises various conditions, including isolated liver steatosis (metabolic dysfunction-associated steatotic liver, MASL), metabolic dysfunction-associated steatohepatitis (MASH, previously non-alcoholic steatohepatitis, NASH), as well as fibrosis, cirrhosis, and MASH-related hepatocellular carcinoma (HCC).

Celiac disease (CD), an autoimmune enteropathy triggered by the intake of gluten, is a widespread pathology, with a worldwide estimated prevalence of approximately 1%, and it is characterized by both gastrointestinal and systemic symptoms, both in adults and children, which are usually resolved by eliminating gluten from the diet (gluten-free diet, GFD) [11]. Although it is often associated with malabsorption symptoms (steatorrhea, weight loss, nutritional deficiencies, etc.), a growing number of sufferers are overweight or frankly obese [12]. One of the conditions that is most frequently detected in paucisymptomatic/asymptomatic subjects is an increase in transaminase levels, which often regresses completely after a GFD is started [13].

A greater risk of NAFLD has emerged in recent years (even though today we should more properly talk about MASLD) in both children and adults with CD adhering to a gluten-free diet (GFD) compared to the general population [14,15] so that the European Society for the Study of CD (ESsCD) has indicated NAFLD as a possible comorbidity of CD [12]. Several pathogenetic mechanisms have been suggested to explain LS development in CD subjects: intestinal mucosa inflammation, increased intestinal permeability (IP), changes in the gut microbiota, and exocrine pancreatic insufficiency [16,17,18,19,20].

Despite the growing amount of evidence, to date, few studies have focused on this condition in the pediatric age group, some reporting that the prevalence of CD among children with NAFLD could be comparable to the prevalence of CD in the general pediatric population [21]. Nevertheless, a study conducted on 11,488 children (0–19 years) affected by CD compared with 57,029 healthy children (of the same age and sex) supports the hypothesis of a close association between CD and NAFLD in childhood [14]. Even less evidence about possible pathogenic mechanisms and/or pre or post-GFD variables associated with MASLD development in CD children has been reported.

In this scenario, we wanted to analyze how this specific condition is characterized, both before and after GFD, in a CD pediatric population attending the main third-level centers for the diagnosis and treatment of CD in western Sicily (Italy). The primary aim of this study was to retrospectively define the prevalence of hypertransaminasemia (and potentially associated HS) in a pediatric population with CD before starting a GFD and to analyze how the introduction of the GFD could modify this condition. The secondary aim was to define the main demographic, clinical, laboratory, and histological features of pediatric CD subjects that correlate with hypertransaminasemia. Finally, in light of these retrospective data, we also conducted a state-of-the-art literature review of the associations between hypertransaminasemia, MASLD, and CD in both children and adults to increase physicians’ awareness and allow an early diagnosis and prompt treatment.

## 2. Materials and Methods

### 2.1. Population and Study Design

This was a retrospective and multicenter study performed on pediatric patients suffering from CD who were consecutively diagnosed according to the European Society for Paediatric Gastroenterology Hepatology and Nutrition (ESPGHAN) criteria [11] between 01/2018 and 10/2023 in three different pediatric departments of Palermo, Italy: Pediatric Gastroenterology Unit, “Di Cristina” Hospital; Department of Pediatrics, University Hospital of Palermo; and Pediatrics Unit, “V. Cervello” Hospital, Ospedali Riuniti “Villa Sofia-Cervello”.

The clinical records of the patients were retrospectively reviewed, and the data obtained were entered into a computerized database.

#### 2.1.1. Population and Study Design

The clinical charts of children aged 1–14 years who were consecutively diagnosed with CD according the ESPGHAN criteria [11] in the three study centers were screened to be included in the study according to the criteria below.

#### 2.1.2. Inclusion Criteria

1.Age > 0.5 and <15 years;2.Complete CD serological panel (anti-gliadin antibody (AGA) immunoglobulin (Ig) A and G, tissue transglutaminase (tTg) IgA and IgG, and anti-endomysial antibodies (EMA));3.Liver transaminase assay performed at diagnosis.

#### 2.1.3. Exclusion Criteria

1.Self-exclusion of gluten from the diet before CD diagnosis;2.Type I and type II refractory CD diagnosis;3.Diagnosis of chronic inflammatory bowel disease or other organic pathologies affecting the digestive system (e.g., wheat allergy, microscopic colitis, diverticulitis, segmental colitis associated with diverticulosis, etc.), neurological diseases, major psychiatric disorders, infectious diseases, immunological deficiencies, and impairments limiting physical activity;4.Incomplete clinical records, lacking the data considered for the present study;5.Lack of clinical follow-up for at least 12 months after diagnosis with >2 outpatient visits during the follow-up period;6.Alcohol intake (no threshold) by the child and/or by the mother during pregnancy or breastfeeding;7.Chronic hepatotropic virus infections (hepatitis B virus (HBV) and hepatitis C virus (HCV));8.Autoimmune liver diseases;9.Congenital metabolic liver diseases (e.g., alpha-1 antitrypsin deficiency, hemochromatosis, Wilson’s disease, porphyria, other storage diseases, etc.);10.Chronic long-term treatment with drugs associated with hepatic damage and liver enzyme modification (e.g., paracetamol, antibiotics, antiepileptics, etc.) [22,23];11.Chronic long-term treatments with drugs associated with both macrovesicular (glucocorticoids, estrogens, tamoxifen, amiodarone, methotrexate, and 5-fluorouracil) and microvesicular (glucocorticoids, valproic acid, tetracycline, and zidovudine) steatosis [24].

In accordance with the ESPGHAN CPGs, the CD diagnosis was performed with the ‘no biopsy approach’ [11]. However, in a subgroup of patients, esophago-gastro-duodenoscopy (EDS) with a biopsy was performed, whenever required by good clinical practice, and the histological findings were categorized according to the Marsh–Oberhuber classification [25,26].

### 2.2. Outcomes

#### 2.2.1. Primary Outcome

To establish the prevalence of hypertransaminasemia not related to other diseases in the CD children, aspartate aminotransferase (AST) and alanine aminotransferase (ALT) data were collected for all the subjects. Data were analyzed both before the start of the GFD and after 1 year of adherence to analyze how the introduction of the GFD could modify this condition. Hypertransaminasemia, differentiated by the age and sex of the children, was defined according to the international validated threshold values [27,28,29].

Accurate anamnesis and specific tests (e.g., study of autoimmune disease, search for genetic hypertransaminasemia causes, etc.) were performed to exclude all possible causes of hypertransaminasemia not related to CD, according to the multi-society CPGs [30].

All the subjects with hypertransaminasemia underwent a liver US examination, both before and after 1 year of the GFD to assess the possible association with HS, according to the methods proposed by Shannon et al. [31] and recently endorsed by multi-society CPGs for pediatric steatotic liver disease [30].

A liver US examination at baseline (i.e., before starting the GFD) was also performed in subjects with atypical CD during the diagnostic workup to exclude possible alternative conditions that might have been responsible for the symptoms reported by the children. The liver US was repeated after 1 year of the GFD in all the patients who had HS at baseline.

#### 2.2.2. Secondary Outcome

To evaluate any conditions that could be associated with hypertransaminasemia, the following baseline (i.e., before the CD diagnosis, on a gluten-containing diet) demographic, clinical, laboratory, and histological (only when EDS was performed, see above) features were analyzed:Sex, age at diagnosis, and ethnicity;Body mass index (BMI) divided into subclasses according to pediatric standards [32];Familial history of CD;CD clinical presentation, including type of CD (typical, atypical, or silent), type of stool disorder (diarrhea, constipation, or mixed bowel movements), abdominal pain, weight loss, growth retardation, evidence of coexisting autoimmune disorders, extraintestinal symptoms, hypertension, and anemia (defined according to age and sex) [33];Complete blood count;Iron metabolism parameters (serum total iron, ferritin, and transferrin levels);Liver function and cholestasis indexes (serum albumin, total and direct bilirubin, alkaline phosphatase (ALP), and gamma glutamyl transpeptidase (GGT) levels, and the international normalized ratio (INR));Non-invasive hepatic fibrosis predictive scores (AST-to-platelet ratio index (APRI) [34] and fibrosis-4 (FIB-4) index [35]);Fasting blood glucose level;Lipid metabolism indexes (total cholesterol, high-density lipoprotein (HDL), low-density lipoprotein (LDL), and triglyceride levels);Thyroid function indexes (thyroid-stimulating hormone (TSH), triiodothyronine (fT3), and thyroxine (fT4) levels);CD autoantibodies (AGA IgA and IgG, tTg IgA and IgG, EMA);HLA status;Marsh histology score (only in patients who underwent EDS during the diagnostic work-up).In addition, the following clinical and laboratory parameters were collected and analyzed after 12 months of the GFD to establish a putative correlation with the disappearance or persistence of hypertransaminasemia:CD symptom modification (asymptomatic at diagnosis, disappearance, reduction, unchanged, or increase; a 0–10 point visual analog scale (VAS) was used due to the differences in symptoms reported at baseline, which included both intestinal and extraintestinal symptoms, and to the lack of a specific validated score for CD symptoms in a pediatric setting; an increase/reduction of ≥2 points was considered a significant modification);GFD adherence score (the 0–4-level Biagi/Pavia score was used) [36];CD autoantibodies (AGA IgA and IgG, tTg IgA and IgG, EMA).

### 2.3. Statistical Analysis

The data are expressed as the means ± standard deviations (SDs) when the distribution was Gaussian, and the Student’s *t* test was used to evaluate differences between groups. Otherwise, the data are expressed as medians and interquartile ranges (IQR) and were analyzed with the Mann–Whitney U tests. Normality was assessed with the Shapiro–Wilk test.

The paired tests were performed with Student’ T test for paired data when the distribution was Gaussian, otherwise with the Wilcoxon test.

The χ^2^ test and Fisher’s exact test were used to compare frequency values in the various population groups, and the McNemar test was used to compare paired proportions.

IBM SPSS Statistics, version 27.0 software (IBM SPSS Inc., Chicago, IL, USA), and MedCalc, version 22.0 (MedCalc Software, Acacialaan, Ostend, Belgium) were used for the statistical analyses.

Values of *p* < 0.05 were statistically significant. All subjects agreed to participate in the study and informed consent forms were signed by the children’s parents or legal guardians. The protocol was approved by the Ethics Committee 2 of Palermo (report n. 806/11_27_2024) and the study was registered on the ClinicalTrials.gov website (protocol n. NCT06206616).

### 2.4. Review of the Literature

For this narrative review, the research group evaluated associations between CD and liver damage, ranging from isolated hypertransaminasemia to liver cirrhosis. The main literature sources were PubMed/MEDLINE and Scopus, and the literature search was conducted in November 2024 by combining free text words and medical subject headings (MeSH). Each keyword was then combined using the Boolean operators “AND” and “OR”.

## 3. Results

The clinical records of 317 children diagnosed with CD between 01/2018 and 10/2023 were reviewed. After the application of the inclusion/exclusion criteria, 160 patients were found eligible and then recruited (Figure 1).

### 3.1. Study Population

Most of the children were female (65.0%) and had a median age of 6.4 (0.8–13.2) years. The majority of them presented with the typical manifestations of CD (50.6%), although only a minority (18.1%) reported diarrhea, but extraintestinal manifestations (25.0%) and anemia (22.5%) were also recorded (see Appendix A). In our study population considered as a whole, no particular abnormalities were found in the main blood chemistry parameters (see Appendix A), with the exception of the obvious positivity for CD antibody markers, as well as the HLA DQ2/DQ8 configuration. Only 26 subjects were required to undergo EDS plus biopsy to confirm the diagnosis (or rule out the coexistence of other conditions), thus the ‘no biopsy approach’ was applied in 83.8% of the children enrolled (see Appendix A).

### 3.2. Hypertransaminasemia and HS Before the Start of the GFD

The prevalences of hypertransaminasemia and HS at diagnosis (i.e., before starting the GFD) were, respectively, 8.1% (n: 13) and 6.1% (n: 6, considering only the 99 patients who underwent a US examination before diagnosis).

An analysis of the baseline features of the patients with hypertransaminasemia compared to all the other CD children showed a higher prevalence of patients with a lower age (*p* = 0.01), the presence of HS (*p* = 0.034), and anti-tTg IgG positivity (*p* = 0.046).

Similarly, in the subjects with HS at the US examination, a higher prevalence was found for those with a lower age (*p* = 0.0001), as well as higher frequency of hypertransaminasemia (*p* = 0.029).

No patients had either an APRI or FIB-4 indicative of a potential evolution towards liver fibrosis development.

The complete data are reported in Table 1, Table 2 and Table 3.

### 3.3. A GFD and Its Effects on Hypertransaminasemia and HS

The data from all the enrolled patients were reassessed after 12 months of the GFD. Overall, dietary adherence was high, with over 90% of subjects showing strict adherence (Biagi score of 3–4), which corresponded to a reduction in symptoms (completely disappearing in 53.1% of cases and reduced in 24.4%), and a significant reduction in CD immunological biomarkers (see Appendix A).

The dietary intervention in the hypertransaminasemia group led to a reduction (or even the complete disappearance of symptoms) in 84.6% of patients, with a perfect correspondence between adherence to the diet and symptom improvement. Similarly, both transaminase levels and CD immunological biomarkers were significantly reduced after 12 months of the GFD, although hypertransaminasemia persisted in 53.8% (n.: 7) of patients (see Table 4 and Table 5). No de novo increases in liver transaminase levels were reported.

A comparison of the main clinical, US, and laboratory features, both at baseline (T0) and after 12 months of the GFD (T12), of patients with hypertransaminasemia persistence to those of non-persistent hypertransaminasemia patients showed no main differences, apart from a greater reduction in liver transaminase levels in the non-persistent hypertransaminasemia group (see Table 6).

Finally, it should be noted that in half of the patients, all belonging to the subgroup without hypertransaminasemia at baseline, there was a complete resolution of HS after 12 months of the GFD, whereas in all three patients with hypertransaminasemia and HS at diagnosis, the presence of HS was confirmed after 12 months of the GFD. However, the small number of patients did not permit reliable correlation analyses.

## 4. Discussion

CD is a chronic autoimmune disorder triggered by the ingestion of the gliadin fraction of gluten, which induces intestinal permeability (IP) alterations in genetically predisposed individuals. It is a very widespread disease, with a prevalence of around 0.5–2% in the general population [37,38,39]. Despite the increase in documented incidence in the last few decades, CD is still underdiagnosed today, with a prevalence of around 0.37% in Italy and 0.35% in Sicily [40].

The pathogenesis of CD is not entirely understood, but it is thought to be due to a combination of genetic, environmental, and immunological factors. The most important genetic susceptibility factors are HLA-DQ2 and HLA-DQ8, whereas gluten is the primary trigger for the development of an immune response in the gut epithelium [41]. The intestinal enzyme tTg2 modifies gluten peptides, which bind to HLA-DQ2 or HLA-DQ8 on the surface of antigen-presenting cells. These trigger a T-cell response, with the release of proinflammatory cytokines, leading to mucosal inflammation and damage to the epithelium. These can induce a B-cell response, leading to the production of anti-tTG antibodies as well. Peptides are capable of directly activating epithelial cells to produce cytokines, such as interleukin (IL)-15. This enhances the cytolytic activity of intraepithelial lymphocytes and ultimately disrupts the lining and increases IP [42].

CD is characterized by gastrointestinal symptoms (diarrhea, abdominal pain, bloating, etc.) and/or systemic symptoms (asthenia, headache, skin manifestations, anemia, hypertransaminasemia, etc.) both in adults and in children [43,44,45,46,47,48,49]; it is nowadays considered a systemic disorder, and is associated with other systemic diseases, such as type 1 diabetes [50], autoimmune thyroiditis, *Sjögren’s* syndrome, psoriasis, microscopic colitis, and dermatitis herpetiformis [51,52]. The pathogenesis behind the extraintestinal manifestations of CD is still not entirely understood, but it is thought to be due to autoantibodies that target and bind Tg2 in intestinal and extraintestinal tissues. Overexpression of Tg2 in the liver, causing IgA antibody deposition, could potentially explain the liver damage in CD patients [53].

### 4.1. Liver Damage in Celiac Disease: From Isolated Hypertransaminasemia to Liver Cirrhosis

CD has been found in up to 9% of patients with elevated liver enzyme levels in the absence of other causes [54,55,56]. Therefore, screening for CD should be part of the workup of patients with an otherwise unexplained increase in liver enzymes levels if other, more common, causes of liver disease have been ruled out [56,57]. If the tests for CD are positive, then a small-bowel biopsy should be performed, in keeping with the guidelines [25,58,59,60].

Conversely, isolated hypertransaminasemia has been observed in 15–55% of asymptomatic or paucisymptomatic CD patients [55,61,62], and in most of them, liver enzymes normalized on a GFD [13]. Bonamico et al. reported increased liver enzyme levels in 60% of CD children [63]; later, many studies detected hypertransaminasemia in 9.2–47.7% of CD pediatric patients [64,65,66,67,68,69]. Only in one study enrolling 149 CD children [70] was hypertransaminasemia lower (6.7%) and it was correlated with lower folic acid levels compared with patients with normal liver enzymes. In a recent Russian study [71], increased liver enzyme levels were observed in 55.9% of CD patients and were correlated with elevated anti-endomysial antibody (EMA) levels. In a recent systematic review and meta-analysis [72] including 42 studies, hypertransaminasemia was identified in 21.42% of CD patients (95% CI: 17.02–26.59, I^2^ = 94%) overall, with a similar prevalence in adults (21.20%) and children (21.51%); 86.4% of them showed a response to the GFD.

“Celiac hepatitis”, the term specifically used to identify liver injury in CD patients that resolves after a GFD [38], is characterized by Kupffer cell hyperplasia on liver histology, with a preserved architecture and a mild mononuclear infiltrate of the portal and lobular tracts [53,73,74]. It ranges from mild to severe hepatitis; intraepithelial lymphocytes can also be seen in the interlobular bile ducts, as well as in the small bowel [62].

The mechanisms underlying the development of abnormal liver enzyme levels in CD patients are still speculative. A predisposition to autoimmunity and the systemic effects of abnormal IP are thought to play pathogenic roles [38].

Gliadin induces an increase in IP and MyD88- dependent zonulin release by binding to the chemokine receptor CXCR3 [75,76,77]; zonulin reversibly regulates IP by modulating intercellular tight junctions. Liver damage might be a consequence of the increased IP, which allows toxins, cytokines, and antigens to reach the liver via the portal circulation and cause liver injury through the release of pro-inflammatory mediators. Toll-like receptors expressed in liver cells (such as Kupffer cells, endothelial cells, dendritic cells, hepatic stellate cells and hepatocytes) can recognize lipopolysaccharides (from Gram-negative bacteria) and mount an immune response [78]. Abnormal IP has been shown in patients with CD and their relatives [79]. In the study by Novacek et al. [62], IP was studied in CD patients before the start of a GFD using the lactulose–mannitol ratio (LaMa) test; interestingly, it was greater in patients with elevated serum liver enzyme levels than in those with normal ones (*p* = 0.0001) and it correlated with AST levels. Similar findings were reported in the study by Cooper B.T. [80], where the LaMa test was significantly higher in CD patients compared to nonceliac participants. Moreover, there was a positive correlation with an improvement in the severity of histology findings after a GFD; patients with a lower LaMa excretion ratio after treatment had mostly grade I histological gradings and higher ratios of the villus height to total mucosal thickness, suggesting an association between histological changes and IP.

Another theory hypothesizes that chronic intestinal mucosal inflammation may be the primary trigger; a similar effect has been proposed in patients with other forms of gastrointestinal inflammation, such as ulcerative colitis [81].

It has been reported that after 12 months on a GFD, the abnormal liver enzyme levels completely normalized in 95–96% of CD patients [55,62]. As there is usually another explanation for the hypertransaminasemia in patients whose liver enzyme levels do not improve on a GFD, other causes must be investigated [82], including viral, autoimmune, and metabolic liver diseases. The study of Ludvigsson J.F. et al. [83] suggested that individuals with CD are at increased risk of liver disease, such as acute and chronic hepatitis, primary sclerosing cholangitis (PSC), fatty liver, liver failure, liver cirrhosis or liver fibrosis and primary biliary cirrhosis (PBC); moreover, prior liver disease was associated with a statistically significant 4- to 6-fold increased risk of later CD. Mounajjed et al. [74] studied thirty patients with CD who had undergone liver biopsy and found nine patients with autoimmune hepatitis (AIH), three with PBC and seven with PSC. CD patients with hypertransaminasemia are usually asymptomatic and may not have any CD manifestations or symptoms [38,84].

The association between CD and PBC seems reasonably well established. Logan et al. [85], Ginn et al. [86], and Neuberger et al. [87] reported cases of patients with concomitant CD and PBC where a GFD alone did not normalize aminotransferase levels. In a well-designed epidemiological study from the UK [88], the prevalence of PBC in 143 CD patients was 3% and the prevalence of CD in 67 patients with PBC was 6%. Similar findings were reported in the study by Dickey et al. [89], where both CD and PBC were present in 7% of 57 patients with PBC, whereas EMAs were found in 11% of patients. Almost all patients had no gastrointestinal symptoms. The prevalence of PBC was reported to be 2- to 3-fold higher in the CD patients, while the CD prevalence in the PBC patients ranged from 3 to 7% [90].

The association between PSC and CD was first noted in 1988 by Hay et al. [91]. More recently, another two studies examined this relationship; in the first [92], CD was found in about 3% of patients with PSC, in the second [93], sixty-one patients with PSC were screened for CD and only one of these patients was positive for EMA.

To evaluate the frequency of CD in patients with autoimmune hepatitis (AIH), Volta et al. [94] tested sera from one hundred eighty-one AIH patients for CD autoantibodies, detecting EMA IgA in eight patients (4%). This prevalence is at least eight times greater than the incidence of CD in the general population. Duodenal biopsy was performed in five of these eight subjects, and CD features were present in all of them. Additionally, Di Biase et al. studied [95] seven children with both known CD on a GFD and AIH with mild fibrosis and necrosis at liver biopsy, who had been treated with steroids and azathioprine for 5 years; aminotransferase levels normalized in all of them at 5 years and there was a significant improvement in liver histology. The prevalence of CD in AIH patients was higher compared to that in the general population and was thought to be around 4–6.4% [78].

Other forms of autoimmune liver disease may also be associated with CD; one case report suggested an association between CD and autoimmune cholangitis [96].

In the 1980s, at least three case reports described various forms of fatty liver in patients with CD [97,98,99]. In all cases, patients had severe HS with hepatomegaly and marked elevations in liver enzyme levels; there was clear evidence of malabsorption secondary to advanced CD in most cases. In all cases, a clinical and biochemical improvement was noted shortly after the start of a GFD, and regression of the fatty infiltration was seen in patients who agreed to a repeat liver biopsy. More convincingly, another two studies showed an association between CD and HS; in a study by Bardella et al. [100] three out of thirteen patients with CD (23%) with elevated liver enzyme levels had steatosis on an ultrasound examination, and in a study by Jacobsen et al. [61], seven of twenty-five CD patients (28%) with abnormal liver enzyme levels who underwent a biopsy had more than 25% steatosis.

Furthermore, Wakim-Fleming et al. [101] tested for CD in 204 patients with biopsy-proven cirrhosis. Five patients were found to be positive for CD on a duodenal biopsy. These patients had cirrhosis secondary to NASH, cryptogenic liver disease, PSC, AIH, and alcoholic liver disease. Four of these patients started a GFD and followed it for 2 years. The biochemical/serological abnormalities and small bowel histology normalized following treatment. Model for end-stage liver disease (MELD) scores improved in three patients.

Moreover, CD has been described in patients with end-stage liver disease (ESLD) [102]. In a study from Finland, four patients with severe liver failure awaiting liver transplantation were discovered to have CD. All patients showed a marked clinical improvement in their liver disease following the introduction of a GFD. The investigators then screened one hundred eighty-five patients from their liver transplant population and found that eight patients (4.3%) had CD. As in the other reports, most of these patients did not have significant gastrointestinal symptoms suggestive of CD. Thus, patients with ESLD of known or unknown causes should probably be screened for CD and, if found to be positive, undergo a confirmatory small-bowel biopsy and start a GFD.

In conclusion, in CD patients, if there is evidence for another cause of liver disease, then the treatment of this specific cause should be commenced (if available) and the GFD should be continued to control CD. However, what effect GFD adherence has on the progression of the underlying liver disease is uncertain [57].

### 4.2. Celiac Disease and MASLD: A Link Between Intestinal Damage and Systemic Manifestations

Historically, liver steatosis has been reported to be a possible manifestation of CD in patients still on a gluten-containing diet [98]; the association between CD and fatty infiltration of the liver is notoriously secondary to rapid weight loss or malabsorption in patients with typical CD, characterized by malabsorption symptoms (steatorrhea, weight loss, nutritional deficiencies, etc.) [37,45,103].

In recent years, the clinical presentation of CD has changed and the typical symptoms, such as diarrhea, nutritional deficits and weight loss, have become less frequent, especially in adult patients [103,104], who are more frequently overweight or obese than underweight at diagnosis [12,105]. In a retrospective study [106], 54.8% of treatment-naive CD patients had a body mass index (BMI) within normal limits, although 8.1% of the remainder were either overweight or obese. Several other studies have made similar observations, and up to 44% of patients have been found to be overweight or obese at the CD diagnosis [107,108,109,110].

Moreover, a close correlation between CD and NAFLD has been observed, so that it is now included among the possible CD comorbidities [12]. This evidence should be considered in the light of data reporting NAFLD as a risk factor for liver cirrhosis and primary liver cancer development [7]. Alarmingly, recent evidence has suggested that NAFLD also increases long-term mortality for cancer, liver disease, and CVD (cardiovascular disease) in children and young adults [111].

Since the revision of the nomenclature from NAFLD to MASLD is a very recent development [5], most of the studies in this review used the first definition (NAFLD); for methodological rigor we will therefore continue to refer to NAFLD, although it would be more correct to adopt the MASLD definition. In addition, several re-examinations of existing cohort studies indicate that NAFLD-related findings can be fully extrapolated to individuals with MASLD. As an example, analyses of data of a large tertiary care NAFLD cohort in the National Health Nutrition Examination Survey (NHANES III) database found a nearly complete overlap between NAFLD and MASLD populations, with only 5.3% of individuals with NAFLD not fulfilling the MASLD criteria; the concordance between NAFLD and MAFLD was excellent, with a 0.968 Cohen’s kappa coefficient (95% CI 0.962–0.973) [112].

The prevalence of CD in NAFLD patients has been reported as 2.2–7.9% [56,113,114] and among these patients, the BMI was often in the normal range [114]. In adults, a CD prevalence of around 2–14% has been observed in NAFLD patients. Only a few studies have analyzed this correlation in a pediatric population, reporting the same prevalence of CD in children with NAFLD and in the general pediatric population [21].

Conversely, little is known about the risk of NAFLD in those already diagnosed with CD. Many studies suggest that CD have a three-fold higher risk of developing NAFLD [14,15,56,98,113,114].

Metabolic risk factors and IP abnormalities are potential triggers for NAFLD development in CD patients. Based on the current evidence, alteration of the gut–liver axis is the main hypothesis supporting an association between CD and NAFLD [115,116,117,118]; in particular, altered IP and small intestinal bacterial overgrowth (SIBO) are common findings [115] in CD patients [76] and have also been described in NAFLD patients [116,117], suggesting a possible pathogenic link [118,119].

Miele et al. [116] demonstrated on duodenal biopsy specimens that NAFLD patients had a lower intensity of duodenal ZO-1 staining, which suggests less intact tight junctions, possibly inducing increased IP, bacterial translocation, and hepatic fat accumulation. The gut–liver axis via the portal system has also been proposed as a potential route for inflammatory cytokines, which may trigger the onset of NASH (now renamed MASH, metabolic dysfunction-associated steatohepatitis). In CD patients, the IP alteration induced by gluten is associated with an inflammatory response [120] and a growing translocation of bacteria via the portal circulation from the intestinal lumen to the liver, where they trigger inflammatory processes through *NF-κB*, activated by the Toll-like receptor 4 (TLR4)–lipopolysaccharide (LPS)–LPS binding protein (LPB) complex, that produces proinflammatory cytokines (tumor necrosis factor (TNF)-α, interleukin (IL)-1β, IL-6, IL-12, and IL-18) [121,122]. Additionally, SIBO is common in CD patients [115] and it can be associated with persistent symptoms in patients adhering to a GFD; intestinal microbiota alterations may modify nutrient absorption, increase mucosal damage [123], and contribute to NAFLD pathogenesis [117]. Levels of endotoxin (i.e., LPS), derived from intestinal Gram-negative microbiota, are elevated in the sera of adults [124] and children [125] with NAFLD, suggesting increased IP in these individuals as well. SIBO is more common among patients with NAFLD than healthy individuals [116] and it is associated with higher TNF-α levels, independent of increases in gut permeability markers [117]. Whether a compromised intestinal barrier is a cause or effect of NAFLD requires additional study, though the prevailing theory is that bacteria-derived endotoxins and related cytokines may serve as a “second hit” in a subgroup of patients with HS, leading to NASH development [126]. This hypothesis is further supported by Kamal et al. [127], whose patients with concomitant NAFLD and CD showed a suboptimal and slower histological intestinal improvement after commencing a GFD compared to patients with CD alone.

In the Italian study by Tortora et al. [119], in treatment-naive CD patients, the prevalence of metabolic syndrome (MetS) was 2%, while in a recent Indian study [128], it was relatively high (11.4%); the difference could be related to the definition of MetS (Adult Treatment Panel III criteria used by Tortora vs. the consensus definition for Asian Indians used in the latter study). Furthermore, both studies demonstrated a high prevalence of fatty liver at the CD diagnosis (18% with HS on ultrasonography in the first study and 14.3% observed on FibroScan^®^ in the second study). In the Indian study, 13% of patients had a BMI in the overweight/obese categories at the CD diagnosis.

In a more recent Italian study [129], MetS was reported in 3.24% of the patients at the CD diagnosis, and HS was reported in 1.7%.

In a retrospective study of 221 newly diagnosed CD patients [130], 65 (29.4%) presented NAFLD at the CD diagnosis, while 32 (14.5%) met the criteria for MAFLD (κ = 0.57). There were no significant differences between NAFLD and MAFLD, except for the higher rate of insulin resistance (IR) of the MAFLD patients (75% vs. 33.8%, *p* < 0.001). At 2 years of follow-up, 46.6% of patients developed NAFLD, while 32.6% had MAFLD (κ = 0.71). MAFLD subjects had higher transaminase levels (*p* = 0.03), LDL cholesterol levels (*p* = 0.04), BMI, waist circumference, and IR than NAFLD patients. MAFLD patients showed higher non-invasive liver fibrosis scores than NAFLD subjects (AST-to-platelet ratio index (APRI) = 1.43 ± 0.56 vs. 0.91 ± 0.62, *p* < 0.001; NAFLD fibrosis score (NFS) = −1.72 ± 1.31 vs. −2.18 ± 1.41, *p* = 0.03; and Fibrosis (FIB)-4 = 1.27 ± 0.77 vs. 1.04 ± 0.74, *p* = 0.04).

In a recent systematic review and meta-analysis [131], the pooled prevalences of fatty liver and MetS in treatment-naive patients with CD were 18.2% (95% CI 8.3–30.8%, n = 1237) and 4.3% (95% CI 2.4–6.7, n = 1239), respectively.

Studies dealing with CD and MASLD in treatment-naive CD patients are summarized in Table 7.

### 4.3. MASLD in CD Patients on a GFD: When Treatment Becomes the Origin of New Problems

CD requires a lifelong GFD [103]. Overall, this diet is considered safe, and it can reduce the long-term risks of serious neoplastic and non-neoplastic complications [137], but it must not be forgotten that it is linked to increased lipid [138,139,140,141,142] and carbohydrate [138,143,144,145] intake, and so many adults and children with CD become overweight after commencing a GFD [109,132,133,146]. Recent evidence has also shown that a GFD, being high in fats and simple sugars, can worsen glucose tolerance and induce MetS and fatty liver (MASLD) in a proportion of CD patients [119,129], potentially increasing their CV (cardiovascular) risk [135].

In a case–control study [138], a total of 98 CD subjects (aged 10–23 years) were matched by age, sex, and BMI with 98 nonceliac participants. A nonconsecutive 3-day food record was completed to assess energy, nutrient, and food intake; compared with the control group, the CD patients reported a significantly higher consumption of added sugar (*p* < 0.001) and total fat (*p* < 0.017).

Several studies of adults and children with CD indicate that obesity/overweight at diagnosis is not unusual and that there is a trend toward the development of overweight/obesity in subjects who comply with a GFD [12,110,134,147], increasing the risks of developing MASLD. According to the World Health Organization (WHO), BMI cut-offs of 25–29.9 kg/m^2^ and >30 kg/m^2^ define overweight and obesity in non-Asians [148]. The presence, duration, and severity of obesity are associated with an increased risk of disease progression in patients with MASLD [148]. An American study [146] confirmed that BMI increases in CD patients after they start a GFD, especially in the GFD-adherent group. The visceral fat distribution, i.e., abdominal obesity, is a major risk factor for cardiometabolic disorders. Waist circumference is a crude index of abdominal obesity and visceral fat accumulation; the current cut-offs of >94 cm in men and >80 cm in women for Caucasians (and adjusted for other ethnicities) are associated with an increased cardiometabolic risk [149] and, consequently, an increased risk of MASLD development.

More recently, two systematic reviews including only studies of adults on GFD on the correlation between overweight/obesity and CD were published. In the first, Nikniaz et al. [150] performed a meta-analysis, reporting a statistically significant increase in BMI during the follow-up period (standard mean difference = 0.26; 95%CI, 0.17–0.35; *p* < 0.001), although the mean BMI remained in the normal-weight category. In the second, Potter et al. [151] focused mainly on the effect of a GFD on CV risk factors and reported an increase in BMI, though patients remained within the normal-weight category.

For the first time in the scientific literature, a recent systematic review with a meta-analysis [136] compared adult and pediatric populations with CD on a GFD, showing the prevalence of normal and underweight BMI to be significantly higher in children and the prevalence of an overweight/obese BMI to be significantly higher in adults; moreover, after commencing a GFD, an increase in the BMI category was significantly more frequent in adults, while the opposite was observed in children. Overall, only 9% of CD patients moved up from an underweight/normal BMI category to an overweight/obese BMI category, whereas 20% changed in the opposite direction, moving from the overweight/obese BMI category to the underweight/normal BMI category. The change of category was significantly higher in the pediatric population than in the adult population. This result agrees with the observation that children are more prone to be symptomatic at disease presentation and therefore are more frequently affected by malabsorption, whereas adults more frequently have an atypical or oligosymptomatic presentation [44]. Moreover, this finding is in line with the WHO’s global assessment of overweight and obesity, which shows an increased prevalence of obesity with increasing age [152]).

Evidence of a higher risk of NAFLD in both children and adults with CD adhering to a GFD compared to the general population emerged in the last decade. In a study based on the Swedish Patient Registry, Reilly et al. [14] compared 11,488 CD children (out of 26,816 CD patients) on a GFD and 57,029 healthy children (out of 130,051 healthy individuals) of the same age and sex, observing in the former an increased risk of NAFLD (HR 4.6; confidence interval 95% = 2.3–9.1; *p* = 0.059). The excess risk value was highest in the first year after the CD diagnosis (HR 13.3, 95% CI = 3.5-50.3, *p* < 0.001), it decreased after 5 years (HR 4.2, 95% CI 2.5–7.2, *p* < 0.001), remaining superimposable after 10–15 years (HR 4.6, 95% CI 2.0–10.6, *p* < 0.001), and persisted through 15 years after the diagnosis (HR 2.5, 95% CI = 1.0–5.9, *p* = 0.043). Later on, in a case–control study [15] with a prospective enrolment of CD adult outpatients on a GFD and healthy controls, more than one-third of the CD patients had concurrent NAFLD, which accounted for a three-fold higher risk compared to the general population (in a binary logistic regression model, the adjusted odds ratio was 2.90, 95% CI: 1.64–5.15, *p* < 0.001); the raw prevalence of NAFLD was 34.7% and 21.8% in the CD and control groups, respectively (*p* = 0.006). Additionally, the relative risk of NAFLD was notably higher in the non-overweight CD patients than in the lean controls, with a difference in the multivariate analysis that accounted for a six-fold increase in risk (adjusted odds ratio = 5.71, 95% CI: 2.3–14.19, *p* < 0.001); CD patients developed NAFLD with fewer metabolic risk factors compared to controls. On the contrary, there were no differences in NAFLD prevalence between the two populations when overweight and obese patients were considered, suggesting that traditional metabolic risk factors may mask the effects of the GFD on these patients.

The association between CD and MASLD, especially in patients on a GFD, has still not been explored in much detail. Children [135] and adults [119,128,153,154] with CD on a GFD may have an increased cardiovascular disease (CVD) risk, which overlaps with those for MASLD.

In a multicenter study enrolling 114 CD children on a GFD for at least one year, Norsa et al. [135] identified three or more concomitant CVD risk factors (BMI, waist circumference, LDL cholesterol levels, triglyceride levels, blood pressure, and insulin resistance) in 14% of patients; the most common were high fasting triglyceride levels (34.8%), elevated blood pressure (29.4%), and high concentrations of calculated LDL cholesterol (24.1%); four children (3.5%) had insulin resistance. The prevalence of borderline LDL cholesterol levels (24%) was higher than at the diagnosis (10%) (*p* = 0.090). Moreover, trends toward increases in overweight (from 8.8% to 11.5%) and obesity (from 5.3% to 8.8%) were also seen in patients on the GFD. These findings suggest that screening for CVD risk factors in CD children both at diagnosis and during follow-up is important, as is the need for dietary counseling targeting CVD prevention.

In the study by Emilsson L. et al. [154], CD patients were at higher risks of CVD (hazard ratio, 1.10; 95% confidence interval [CI], 1.03–1.28) and stroke (odds ratio, 1.11; 95% CI, 1.02–1.20).

In an observational prospective study including 98 consecutive adult CD patients (>18 years) diagnosed at the Gastrointestinal Unit of the University of Naples, Tortora et al. [119] observed that CD patients had an increased prevalence of MetS 1 year after starting a GFD (from 2% at diagnosis to 29.5% after 1 year of the GFD) and of HS on US (from 18% at diagnosis to 28.5% after 1 year of the GFD); moreover, 2/98 patients (2%) fulfilled the diagnostic criteria for MetS at diagnosis [155] compared to 29/98 patients (29.5%) after 12 months of the GFD (*p* < 0.01; OR: 20). With regard to the MetS sub-categories, the authors reported the following data 1 year after patients started the GFD compared to the baseline, respectively: in 72 vs. 48 patients the waist circumference cut-off was exceeded (*p* < 0.01; OR: 2.8); 18 vs. 4 patients had high blood pressure (*p* < 0.01; OR: 5.2); 25 vs. 7 patients exceeded the glycemic threshold (*p* = 0.01; OR: 4.4); 34 vs. 32 patients had reduced levels of HDL cholesterol (*p* = 0.7); and 16 vs. 7 patients had high levels of triglycerides (*p* = 0.05). In this study, many CD patients had a high or normal BMI at diagnosis (mean value 22.9 ± 4) but this value increased after the start of the GFD 24.1 ± 4; *p* = 0.01). Moreover, of the whole cohort of 98 patients, 18 (18%) showed HS on US at the CD diagnosis, rising to 28 after 1 year of the GFD (18% vs. 28.5%; *p* = 0.1). HS was present in 19 of the 29 patients with MetS and in 9 of the 69 patients without MetS (65% vs. 13%; *p* < 0.01; OR: 19), suggesting that patients with MetS had a higher risk of excessive fat accumulation in the liver than patients without.

This finding confirms the experience of other authors who demonstrated that HS is also strongly associated with MetS [156,157,158] in CD patients.

In an Italian study [129], adult CD patients showed an increased risk of developing both MetS and HS after following a GFD. MetS was reported in 3.24% of cases at the CD diagnosis and in 14.6% after a GFD (*p* < 0.0001). HS was reported in 1.7% at diagnosis and in 11.1% after starting a GFD (*p* < 0.0001). Regarding the metabolic sub-categories, the prevalences of increased waist circumference, hypertension, reduced HDL cholesterol levels, hyperglycemia, hypercholesterolemia, and a BMI > 25 were significantly higher after starting a GFD compared to the baseline values at the CD diagnosis (*p* = 0.0001 for each parameter).

In a recent Indian study [128], the prevalences of MetS and fatty liver were assessed in two cohorts of CD patients. After 1 year of a GFD, MetS prevalence increased from 11.4% at diagnosis to 18.2% (*p* = 0.219) and the prevalence of fatty liver rose from 14.3% at diagnosis to 29.5% (*p* = 0.002). The prevalence of HS in the CD patients who had been following a GFD for more than 1 year was 23%. Moreover, the proportion of patients with BMI in the overweight/obese categories increased to approximately 25% after 1 year of a GFD. Interestingly, the increase in BMI was mainly in body fat rather than in muscle mass, as suggested by the increases in body fat percentage and BMI. Among the individual parameters of the MetS after 1 year of a GFD: the number of patients with hyperglycemia increased from five (11.9%) to thirteen (30.9%) (*p* = 0.039) and the number with hypertriglyceridemia increased from four (9.1%) to seven (15.9%) (*p* = 0.375); one patient developed hypertension. A slight decrease in the proportion of patients with low HDL levels was reported (before the GFD 74.4% vs. after 63.6%; *p* = 0.392), mainly in male patients; there was also a significant rise in body weight (before 49.0 ± 13.7 kg vs. after 53.2 ± 13.3 kg; *p* < 0.001), thus resulting in a significant increase in BMI (*p* = 0.003). The percentage of overweight and obese patients increased from 13.6% at baseline to 25% after 1 year of a GFD. There was also an increase in the number of patients with an elevated waist circumference (before 22.7% vs. after 31.8%; *p* = 0.125), with increases in both the mean waist circumference (before 76.9 ± 10.9 cm vs. after 80.1 ± 12.5 cm; *p* = 0.003) and the mean hip circumference (before 84.6 ± 7.9 cm vs. 88.0 ± 8.7 cm after; *p* < 0.001). The number of patients with fatty liver increased from six (14.3%) at baseline to thirteen (29.5%) (*p* = 0.002), with a significant increase in mean CAP values from 213.5 ± 46.9 dB to 230.8 ± 50.1 dB (*p* = 0.001), while no significant differences in liver stiffness were observed (*p* = 0.098). Furthermore, there was an increase in the proportion of patients with a more severe degree of fatty liver, with eight patients (18.2%) having grade III fatty liver after 1 year of a GFD compared to one (2.4%) at baseline.

In a recent meta-analysis of 11 cross-sectional and longitudinal studies (analyzing data from 2578 subjects) [131], the pooled prevalence of fatty liver and MetS in CD patients on a GFD for varying durations (from 6 months to 36 years) was 28.2% (95% CI 20.7–36.4%, n = 1368) and 21.3% (95% CI 11.7–32.9%, n = 2193), respectively, compared to 18.2% (95% CI 8.3–30.8%) and 4.3% (95% CI 2.4–6.7) before the start of the GFD.

Studies dealing with CD and MASLD in patients on a GDF are summarized in Table 8.

Despite the emerging evidence that a sizeable proportion of CD patients on a GFD gain weight and eventually develop MetS, the pathophysiological mechanisms of this condition are not clear. Persistent gut–liver axis alterations and an unfavorable composition of the GFD have been proposed as potential factors [119]. To support the first hypothesis, a phenomenon of hyperabsorption has been described after the restoration of intestinal mucosal function with the GFD in CD patients [159]; a differential secretion pattern of brain–gut axis hormones has also been described in patients with both untreated and treated CD, which may underlie the weight gain and development of MetS [159,160]. For the second, the nutritional composition of packaged gluten-free foods (PGFF) has been proposed as a possible cause of overweight/obesity [161]. Several studies have confirmed that a long-term GFD may not be nutritionally balanced [14,161]; a GFD is high in fat and sugar and low in fiber (pasta and cereal-based food, sweet snacks, cookies, cakes, and bread and bakery products) and induces a higher intake of simple sugars with a higher glycemic index and higher protein and saturated fat contents, and the intake of complex carbohydrates [162,163,164]. Many gluten-free foods are characterized by a glycemic index that is thought to be higher than that of equivalent gluten-containing foods [165,166], although this report has been refuted by some authors [167]; the higher glycemic index of gluten-free products could be partially explained by the impairment caused by gluten to the amylase-mediated hydrolysis of starch granules in the lumen of the small intestine (reduction of ‘accessibility’), thus decreasing starch adsorption. Furthermore, the high quantity of saturated fat used in preparing processed food for a GFD to make it palatable also adds to the higher calorie intake. Finally, the unpalatability of some gluten-free foods may induce a preference towards hyperproteic and hyperlipidemic foods [162,168], which, in turn, may lead to an increased energy intake and subsequent weight gain.

Moreover, the association between overweight/obesity and insulin resistance in CD patients on a GFD has been explained by two immunological/inflammatory hypotheses. First, tumor necrosis factor alpha (TNFα) expression is increased in obese humans and is thought to participate in insulin resistance mechanisms by inhibiting tyrosine kinase activity at the insulin receptor; therefore, the increased BMI during a GFD might explain the higher prevalence of insulin resistance in CD patients at follow-up [169,170,171]. According to the second hypothesis, which is supported by the study of Luciani et al. [172], tTG drives inflammation in CD patients via the downregulation of peroxisome proliferator-activated receptor gamma (PPARG). As PPARG upregulation is involved in type 2 diabetes susceptibility [173], it is possible that by reducing inflammation, the GFD might also influence this pathway. However, this theory needs to be corroborated by further research.

Figure 2 summarizes the possible mechanisms that might be involved in MASLD development in CD patients both before and after the introduction of the GFD.

### 4.4. Another Small Piece Toward Completing the Puzzle: Hypertransaminasemia and HS in a Cohort of Sicilian Children with Celiac Disease

In light of the data reported in the international literature, but above all of the gaps in our understanding that are still present today, particularly in the pediatric population, we decided to investigate the prevalence of hypertransaminasemia and HS in children presenting to the main referral centers for CD in western Sicily.

In the 160 children with CD enrolled, prevalence values of 8.1% and 6.1% were observed for hypertransaminasemia and HS, respectively.

As regards hypertransaminasemia, the data for our population differed significantly from many of the data reported in the literature, which attest prevalence rates between 9.2% and 47.7% in various geographical areas (from the USA to Eastern Europe and the Middle East) [64,65,66,67,68,69], but which, overall, seem to indicate that approximately 21% of children with CD have hypertransaminasemia [72]. However, not all the international data differ from what we have highlighted. In a Polish study, which was also conducted retrospectively and on a population similar to ours in terms of numbers, sex, and average age, the prevalence of increased transaminase levels was 6.7% and evidence of HS was 8.1% [70]. In that study, the authors identified a positive correlation between IgA anti-TG2 concentrations and serum ALT activity, whereas our data proved just a small but significantly higher frequency of anti-tTg IgG positivity in subjects with hypertransaminasemia. Similarly to other studies, which, however, had a higher prevalence of hypertransaminasemia, in our pediatric population, children with increased levels of liver function-related enzymes presented at an earlier age [64,66]. Of note is the close correlation between high transaminase values and US evidence of HS (23.1% in patients with hypertransaminasemia vs. 3.5% in patients without hypertransaminasemia), as well as the absence of any association with the classic risk factors of MetS (e.g., obesity, hypertriglyceridemia, etc.). This fully reflects what has already been reported in many other studies on children with celiac disease [64,65,66,72], supporting the hypothesis that in pediatric patients, metabolic factors play a minor/irrelevant part in liver disorders associated with CD (unlike in adults, where their role is more significant), leaving the main role to the immunological factors linked to the increase in IP and the translocation of enteric products with consequent immunogenic activity [76,115,116,117,118,119].

As regards US evidence of HS, although the small number of our patients is a non-negligible factor and that the US method is not the gold standard for the study of this condition, particularly in children, our study is one of the very few [64,68,70] to report the prevalence of this condition in children with celiac disease before starting a GFD, setting it between 5% and 8%. This result differs significantly from those found in adults, where it is estimated that approximately 18% of celiac patients already have HS at diagnosis [131]. Similarly to what has already been reported for hypertransaminasemia, there were correlations between HS and the lower age group and the presence of hypertransaminasemia, while no specific associations were found with the clinical, biochemical, or immunological features that characterize MetS, further strengthening the hypothesis of the low relevance of metabolic factors as a cause of hepatic involvement in children with celiac disease.

A 12-month GFD intake resulted in mainly clinical benefits in patients with hypertransaminasemia, with a reduction or disappearance of symptoms in over 80%, but transaminase levels normalized in less than 50%, and steatosis did not disappear in any of the patients with both hypertransaminasemia and HS (vs. a complete regression in patients in whom HS was not associated with hypertransaminasemia). The persistence of hypertransaminasemia in our group of patients seems to be in contrast with other studies [65,66,67,68], such as, for example, the study by Farre et al. [64], in which transaminase levels normalized in 100% of pediatric patients with CD following a GFD, either before or at the time of negativization of the CD immunological markers. In our analysis, no significant differences were identified either at baseline or after 12 months of a GFD in patients with or without normalized transaminase levels. However, it should be emphasized that our follow-up stopped at 12 months of a GFD and that, therefore, longer periods of the diet could have led to a complete regression of the liver involvement.

Finally, although the value of the predictive scores for the risk of developing liver fibrosis (i.e., APRI and FIB-4) has been recognized by the main scientific hepatology societies as being severely limited in the pediatric population, we believe it is relevant to mention that none of the subjects, whether with or without hypertransaminasemia or with or without HS, presented high scores. This finding, which potentially minimizes the risk of evolution toward more severe forms of liver involvement in children with CD, must be confirmed with scores specific for the pediatric age group and using elastography methods; neither of these were included in our study due to its retrospective nature.

This study suffers from some limitations that need to be mentioned. The retrospective design of the study made it impossible to obtain multiple data that could have clarified the clinical and etiopathogenetic aspects of the conditions analyzed. In particular, the absence of specific biomarkers to evaluate intestinal bacterial growth, IP, tissue damage, and immunological activation did not allow us to draw conclusions on the pathophysiological mechanisms involved, but only to hypothesize possible explanations consistent with reports already existing in the literature. Similarly, and again due to the retrospective nature of the study, the most appropriate methods for analyzing both HS and hypothetical fibrotic damage to the liver were not used. A further limitation is related to the small number of patients, who were enrolled at third-level centers, which could have created an implicit selection bias by including in the analysis patients with more complex clinical and immunological features, and, therefore, the results may not be extendable to the entire population of children with celiac disease. Finally, the limited follow-up period considered in this study does not allow us to exclude the possibility that longer periods of a GFD may improve patient outcomes or that new elements, not considered in our analysis, may emerge.

## 5. Conclusions

To date, a GFD is the only available treatment for CD; even if it is considered a quite safe therapeutic option, it is associated with increases in overweight/obesity, MetS, and HS (MASLD). Most CD patients start a GFD at a young age, and so a long-term GFD can lead to a perpetual and progressive worsening of metabolic disorders, predisposing CD patients to a higher CVD risk.

In the last few decades, many studies have suggested high risks of overweight/obesity and NAFLD in CD patients, which increase after a GFD is started. A few emerging studies have also suggested an increased risk of MASLD in CD patients on a GFD; therefore, the nutritional and metabolic features of these patients should be assessed at regular intervals.

However, the liver involvement in CD patients is not solely secondary to GFD intake, but already exists before the CD diagnosis, with prevalence rates differing in children and adults. Only a few conflicting works have analyzed hypertransaminasemia and HS in pediatric CD patients. Our study found prevalences of 8.1% and 6.1% for hypertransaminasemia and HS, respectively, which were below the values reported in the literature, although these seem to confirm some of the clinical features (a greater association with a lower age at diagnosis already reported by other authors). Of note, the absence of the typical parameters of MetS in these patients seems to exclude the metabolic origin of these disorders in a pre-GFD state, while giving prominence to the immunological one. A GFD can reverse the trend of liver involvement, although periods longer than 12 months may be necessary.

Consequently, appropriate preventive strategies should be initiated, including early screening and counselling to promote physical activities and a well-balanced GFD. CD patients should be advised to avoid adding excess fats and sugars to homemade GFD products and avoid buying commercially available products high in fats and sugars. All medical units diagnosing CD should contemplate including in their staff a nutritionist with experience in this particular clinical field. Dietary advice provided using a patient-tailored approach should assist CD/MASLD/HS patients in achieving an appropriate nutritional intake whilst reducing the risk of long-term liver-related events. It is also important for health authorities to legislate on and monitor the calories and fat contained in gluten-free products.

## Figures and Tables

**Figure 1 nutrients-17-00085-f001:**
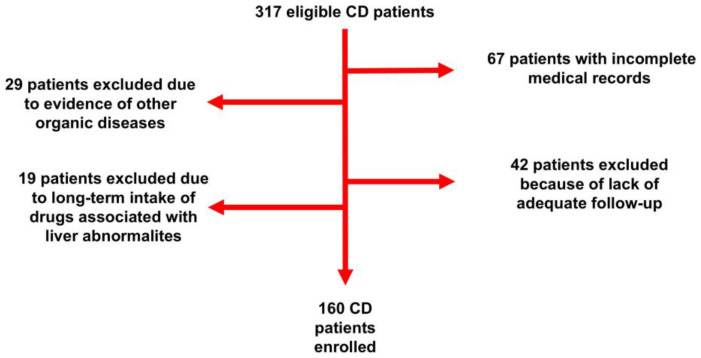
Flow chart of the study. CD: Celiac disease.

**Figure 2 nutrients-17-00085-f002:**
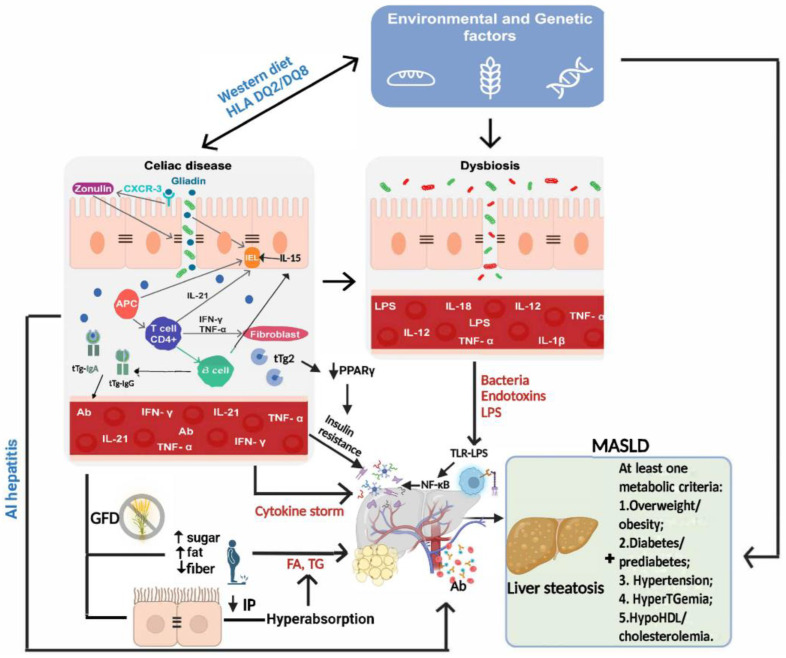
Possible pathogenic links between celiac disease and MASLD development both before and after the GFD introduction. AI: autoimmune; Ab: antibodies; APC: antigen-presenting cell; CD: celiac disease; CXCR: C-X-C motif chemokine receptor; FA: fatty acids; HDL: high-density lipoproteins; IEL: intraepithelial lymphocytes; IFN: interferon; Ig: immunoglobulin; IL: interleukin; IP: intestinal permeability; LPS: lipopolysaccharide; MASLD: metabolic dysfunction-associated steatotic liver disease; NF-kB: nuclear factor kappa-light-chain-enhancer of activated B cells; TNF: tumor necrosis factor; TG: triglyceride; TLR: Toll-like receptor; tTg: tissue transglutaminase.

**Table 1 nutrients-17-00085-t001:** Baseline (T0) demographic and clinical features of the CD population according to hypertransaminasemia or US liver steatosis evidence.

	No Hypertransaminasemia (N = 147)	Hypertransaminasemia (N = 13)	*p*	No Liver Steatosis on US (N = 92)	Liver Steatosis on US (N = 6)	*p*
Age at CD diagnosis (mean ± SD)						
0–1.5 years (n, %)	5 (3.4)	3 (23.0)		1 (1.1)	3 (50.0)	
1.5–5 years (n, %)	68 (46.3)	5 (38.5)	0.01	49 (53.3)	1 (16.7)	0.0001
6–14 years (n, %)	74 (50.3)	5 (38.5)		42 (45.6)	2 (33.3)	
Sex:						
Male (n, %)	96 (65.3)	5 (38.5)	NS	35 (38.0)	2 (33.3)	NS
Female (n, %)	51 (34.7)	8 (61.5)		57 (62.0)	4 (66.7)	
Ethnicity:						
Caucasian (n, %)	146 (99.3)	13 (100.0)	NS	91 (98.9)	6 (100.0)	NS
African (n, %)	1 (0.7)	0 (0.0)		1 (1.1)	0 (0.0)	
BMI class:						
Underweight (n, %)	19 (12.9)	3 (23.1)		13 (14.1)	1 (16.7)	
Healthy weight (n, %)	83 (56.5)	4 (30.8)		56 (60.9)	1 (16.7)	
Overweight (n, %)	41 (27.9)	6 (46.2)	NS	20 (21.7)	4 (66.7)	NS
Obesity (n, %)	2 (1.4)	0 (0.0)		1 (1.1)	0 (0.0)	
Severe obesity (n, %)	2 (1.4)	0 (0.0)		2 (2.2)	0 (0.0)	
Familial history of CD (n, %)	23 (15.6)	3 (23.1)	NS	24 (26.1)	1 (16.7)	NS
CD type:						
Typical (n, %)	74 (50.3)	7 (53.8)		19 (20.7)	3 (50.0)	
Atypical (n, %)	46 (31.3)	6 (46.2)	NS	50 (54.3)	1 (16.7)	NS
Silent (n, %)	27 (18.4)	0 (0.0)		23 (25.0)	2 (33.3)	
Stool disorders:						
None (n, %)	91 (61.9)	5 (38.5)	NS	62 (67.4)	4 (66.7)	NS
Diarrhea (n, %)	23 (15.6)	6 (46.2)		9 (9.8)	2 (33.3)	
Constipation (n, %)	17 (11.6)	1 (7.7)		12 (13.0)	0 (0.0)	
Mixed bowel movements (n, %)	16 (10.9)	1 (7.7)		9 (9.8)	0 (0.0)	
Abdominal pain (n, %)	53 (36.1)	4 (30.8)	NS	16 (17.4)	1 (16.7)	NS
Weight loss (n, %)	30 (20.4)	3 (23.1)	NS	8 (8.7)	2 (33.3)	NS
Growth retardation (n, %)	47 (32.0)	5 (38.5)	NS	25 (27.2)	2 (33.3)	NS
Autoimmune disease:						
None (n, %)	120 (81.6)	12 (92.3)	NS	75 (81.5)	4 (66.7)	NS
Type 1 diabetes (n, %)	14 (9.5)	0 (0.0)		10 (10.9)	0 (0.0)	
Autoimmune thyroiditis (n, %)	5 (3.4)	1 (7.7)		3 (3.3)	1 (16.7)	
Type 1 diabetes and autoimmune thyroiditis (n, %)	5 (3.4)	0 (0.0)		4 (4.3)	1 (16.7)	
Others (n, %)	3 (2.0)	0 (0.0)		0 (0.0)	0 (0.0)	
Extraintestinal symptoms (n, %)	37 (25.2)	3 (23.1)	NS	26 (28.3)	1 (16.7)	NS
Hypertension (n, %)	1 (0.7)	0 (0.0)	NS	1 (1.1)	0 (0.0)	NS
Anemia (n, %)	32 (21.8)	4 (30.8)	NS	22 (23.9)	0 (0.0)	NS

CD: celiac disease; NS: not significant.

**Table 2 nutrients-17-00085-t002:** Baseline (T0) laboratory, US, and histological features of the CD population according to hypertransaminasemia or US liver steatosis evidence.

	No Hypertransaminasemia (N = 147)	Hypertransaminasemia (N = 13)	*p*	No Liver Steatosis on US (N = 92)	Liver Steatosis on US (N = 6)	*p*
Hb (g/dL) (mean ± SD)	12.1 (1.5)	11.9 (1.4)	NS	12.18 (1.29)	12.6 (1.58)	NS
MCV (fL) (mean ± SD)	73.6 (8.0)	72.1 (9.6)	NS	73.65 (8.08)	72.65 (9.01)	NS
MCH (pg) (mean ± SD)	24.8 (4.0)	25.9 (5.1)	NS	25.06 (3.42)	28.18 (4.02)	NS
RDW (%) (mean ± SD)	15.9 (5.1)	16.9 (5.5)	NS	16.27 (5.59)	14.6 (1.14)	NS
Total iron (µg/dL) (mean ± SD)	62 (31.6)	64.3 (43.2)	NS	56.6 (30.3)	83.3 (51.6)	NS
Ferritin (ng/mL) (median; IQR)	20.8 (5.8–28.8)	7.8 (3–18.5)	NS	16.73 (4.0–24.0)	34.3 (4.3–69.3)	NS
Transferrin (mg/dL) (median; IQR)	323 (269.5–404.0)	319.5 (258.9–396.2)	NS	375 (225.5–443.5)	356.5 (205.3–398.2)	NS
WBC (10^3^) (mean ± SD)	8.1 (3.9)	9.3 (3.2)	NS	8.5 (4.6)	10.2 (1.4)	NS
Platelets (10^3^) (median; IQR)	306 (265–381)	355 (307.0–448.0)	NS	325.5 (275.2–384.4)	381.5 (245.6–447.0)	NS
AST (U/L) (mean ± SD)	24.7 (6.8)	51.9 (9.9)	<0.0001	27.7 (9.7)	43.0 (25.0)	NS
ALT (U/L) (mean ± SD)	19.9 (6.4)	36.8 (11.2)	<0.0001	22.1 (8.43)	27.8 (17.92)	NS
Hypertransaminasemia (n, %)	NA	NA	NA	10 (10.9)	3 (50.0)	0.029
Liver steatosis on US (n, %)	3/85 (3.5)	3 (23.1)	0.034	NA	NA	NA
Albumin (g/dL) (mean ± SD)	4.3 (0.5)	3.9 (0.5)	NS	4.1 (0.4)	3.9 (0.5)	NS
Total bilirubin (mg/dL) (mean ± SD)	0.3 (0.2)	0.2 (0.3)	NS	0.3 (0.2)	0.1 (0.2)	NS
Direct bilirubin (mg/dL) (mean ± SD)	0.1 (0.1)	0.2 (0.1)	NS	0.12 (0.1)	0.15 (0.2)	NS
ALP (U/L) (mean ± SD)	199.2 (100.5)	200.4 (50.5)	NS	188.2 (59.5)	179.3 (56.2)	NS
GGT (U/L) (mean ± SD)	7.7 (2.5)	13.7 (11.0)	NS	9.5 (5.1)	8.4 (3.6)	NS
Total cholesterol (mg/dL) (mean ± SD)	158.2 (25.3)	145 (24.6)	NS	159.25 (26.5)	144 (23.6)	NS
HDL (mg/dL) (mean ± SD)	56.6 (24.6)	52.4 (23.2)	NS	56.82 (17.6)	54.3 (18.9)	NS
LDL (mg/dL) (mean ± SD)	79.6 (28.5)	82.6 (26.3)	NS	80.07 (26.7)	81.1 (24.3)	NS
Triglycerides (mg/dL) (median; IQR)	69.3 (43.5–101.5)	71.2 (42.2–111.3)	NS	65.5 (46.8–107.5)	69.5 (46.6–117.3)	NS
Fasting blood glucose (mg/dL) (mean ± SD)	111.3 (71.4)	114.2 (72.1)	NS	113.3 (74.3)	118.0 (94.9)	NS
IgA (g/dL) (mean ± SD)	139.0 (72.9)	122.3 (68.9)	NS	118.28 (62.3)	117.4 (63.2)	NS
INR (mean ± SD)	0.9 (0.4)	0.5 (0.7)	NS	0.94 (0.4)	0.95 (0.26)	NS
TSH (U/mL) (mean ± SD)	2.2 (1.1)	3.2 (2.7)	NS	2.32 (1.4)	3.42 (2.7)	NS
fT3 (pg/mL) (mean ± SD)	2.8 (1.7)	2.9 (1.6)	NS	3.1 (1.8)	2.9 (1.8)	NS
fT4 (ng/dL) (mean ± SD)	1.9 (3.1)	0.8 (0.6)	NS	1.17 (0.4)	1.26 (0.1)	NS

ALP: alkaline phosphatase; ALT: alanine aminotransferase; AST-to-platelet ratio index; AST: aspartate aminotransferase; CD: celiac disease; fT3: triiodothyronine; fT4: thyroxine; GGT: gamma glutamyl transpeptidase; Hb: hemoglobin; HDL: high-density lipoprotein; Ig: immunoglobulin; INR: international normalized ratio; IQR: interquartile range; LDL: low-density lipoprotein; MCH: mean corpuscular hemoglobin; MCV: mean corpuscular volume; NA: not applicable; NS: not significant; RDW: red cell distribution width; SD: standard deviation; TSH: thyroid-stimulating hormone; WBC: white blood cell.

**Table 3 nutrients-17-00085-t003:** Baseline (T0) CD immunological and histological features according to hypertransaminasemia or US liver steatosis evidence.

	No Hypertransaminasemia (N = 147)	Hypertransaminasemia (N = 13)	*p*	No Liver Steatosis on US (N = 92)	Liver Steatosis on US (N = 6)	*p*
AGA IgA (U/mL) (median; IQR)	33.6 (4.0–37.0)	265 (4.25–304.0)	NS	32.45 (3.0–36.1)	62.75 (3.0–85.1)	NS
Positive for AGA IgA (n, %)	100 (68.0)	9 (69.2)	NS	60 (65.2)	6 (100.0)	NS
AGA IgG (U/mL) (median; IQR)	36.5 (5.0–52.0)	219.3 (4.5–233.0)	NS	45.9 (4.5–55.5)	80.6 (5.0–89.0)	NS
Positive for AGA IgG (n, %)	109 (74.1)	11 (84.6)	NS	67 (72.8)	6 (100.0)	NS
tTg-IgA (U/mL) (median; IQR)	927.6 (7.0–2700.8)	9062.4 (4.0–13,665.0)	NS	1062.8 (6.0–2947.5)	6804.5 (3.0–12,808.5)	NS
Positive for tTg-IgA (n, %)	145 (98.6)	13 (100.0)	NS	91 (98.9)	6 (100.0)	NS
tTg-IgG (U/mL) (median; IQR)	44.1 (0.4–3500.0)	137.9 (7.0–327.0)	NS	75.45 (5.8–116.0)	98.3 (5.2–111.5)	NS
Positive for tTg-IgG (n, %)	98 (66.7)	12 (92.3)	0.046	66 (71.7)	4 (66.7)	NS
Positive for EMA (n, %)	147 (100.0)	12 (92.3)	NS	91 (98.9)	6 (100.0)	NS
HLA status: DQ2-positive (n, %) DQ8-positive (n, %)	133 (90.5) 14 (9.5)	13 (100.0) 0 (0.0)	NS	85 (92.4) 7 (7.6)	6 (100.0) 0 (0.0)	NS
Marsh score:						
3A (n, %)	8/24 (33.3)	0/2 (0.0)		5/14 (35.7)	0/2 (0.0)	
3B (n, %)	7/24 (29.2)	1/2 (50.0)	NS	3/14 (21.4)	0/2 (0.0)	NS
3C (n, %)	9/24 (37.5)	1/2 (50.0)		6/14 (42.9)	0/2 (0.0)	

AGA: anti-gliadin antibody; CD: celiac disease; EMA: anti-endomysial antibody; Ig: immunoglobulin; IQR: interquartile range; NS: not significant; tTg: tissue transglutaminase.

**Table 4 nutrients-17-00085-t004:** Comparison of clinical, laboratory, and US features at baseline (T0) and follow-up (T12) in CD patients with hypertransaminasemia.

	T0 (N = 13)	T12 (N = 13)	*p*
Symptoms			
Asymptomatic at diagnosis (n, %)		0 (0.0)	
Disappearance (n, %)		5 (38.5)	
Reduction (n, %)	NA	6 (46.1)	NA
Unchanged (n, %)		1 (7.7)	
Increased (n, %)		1 (7.7)	
Biagi/Pavia GFD adherence score:			
Score 0 (n, %)		1 (7.7)	
Score 1 (n, %)		0 (0.0)	
Score 2 (n, %)	NA	1 (7.7)	NA
Score 3 (n, %)		3 (23.1)	
Score 4 (n, %)		8 (61.5)	
AST (U/L) (mean ± SD)	51.9 (9.9)	24.1 (10.2)	<0.0001
ALT (U/L) (mean ± SD)	36.8 (11.2)	19.4 (8.3)	<0.0001
AGA IgA (U/mL) (median; IQR)	265 (4.25–304.0)	15.6 (5.3–18.8)	<0.0001
AGA IgG (U/mL) (median; IQR)	219.3 (4.5–233.0)	17.2 (3.0–20.8)	<0.0001
tTg-IgA (U/mL) (median; IQR)	9062.4 (4.0–13,665.0)	231.2 (3.0–503.0)	<0.0001
Positive for tTg-IgA (n, %)	13 (100)	10 (76.9)	NS
tTg-IgG (U/mL) (median; IQR)	137.9 (7.0–327.0)	13.6 (3.75–19.75)	<0.0001

AGA: anti-gliadin antibody; ALT: alanine aminotransferase; AST: aspartate aminotransferase; CD: celiac disease; GFD: gluten-free diet; Ig: immunoglobulin; IQR: interquartile range; NA: not applicable; NS: not significant; SD: standard deviation; tTg: tissue transglutaminase.

**Table 5 nutrients-17-00085-t005:** Comparison of US and laboratory features at baseline (T0) and follow-up (T12) in CD patients with hypertransaminasemia.

		Liver Steatosis on US at T12			*p*
		Negative	Positive	Tot	
Liver steatosis on US at T0	Negative	10	0	10	
	Positive	0	3	3	
Tot		10	3	13	NA *
		AGA IgA T12			
		Negative	Positive	Tot	
AGA IgA T0	Negative	4	0	4	
	Positive	3	6	9	
Tot		7	6	13	NS
		AGA IgG T12			
		Negative	Positive	Tot	
AGA IgG T0	Negative	2	0	2	
	Positive	5	6	11	
Tot		7	6	13	NS
		tTg-IgA T12			
		Negative	Positive	Tot	
tTg-IgA T0	Negative	0	0	0	
	Positive	3	10	13	
Tot		3	10	13	NS
		tTg-IgG T12			
		Negative	Positive	Tot	
tTg-IgG T0	Negative	1	0	1	
	Positive	6	6	12	
Tot		7	6	13	0.031
		EMA T12			
		Negative	Positive	Tot	
EMA T0	Negative	1	0	1	
	Positive	4	8	12	
Tot		5	8	13	NS

* NA due to the low number of subjects included in the analysis. AGA: anti-gliadin antibody; CD: celiac disease; EMA: anti-endomysial antibody; Ig: immunoglobulin; NS: not significant; tTg: tissue transglutaminase.

**Table 6 nutrients-17-00085-t006:** Comparison of clinical, laboratory, and US features at baseline (T0) and follow-up (T12) in CD patients with persistent vs. non-persistent hypertransaminasemia.

	T0			T12		
	Persistent HT (N = 7)	Non-Persistent HT (N = 6)	*p*	Persistent HT (N = 7)	Non-Persistent HT (N = 6)	*p*
Symptoms						
Asymptomatic at diagnosis (n, %)				0 (0.0)	0 (0.0)	
Disappearance (n, %)				3 (42.9)	2 (33.3)	
Reduction (n, %)	NA	NA	NA	4 (57.1)	2 (33.3)	NS
Unchanged (n, %)				0 (0.0)	1 (16.7)	
Increased (n, %)				0 (0.0)	1 (16.7)	
Adherence (Biagi/Pavia) score:						
Score 0 (n, %)				1 (14.3)	0 (0.0)	
Score 1 (n, %)				0 (0.0)	0 (0.0)	
Score 2 (n, %)	NA	NA	NA	0 (0.0)	1 (16.7)	NS
Score 3 (n, %)				2 (28.6)	2 (33.3)	
Score 4 (n, %)				4 (57.1)	3 (50.0)	
US liver steatosis (n, %)	2 (28.6)	1 (16.7)	NS	2 (28.6)	1 (16.7)	NS
AST (U/L) (mean ± SD)	53.0 (9.5)	50.7 (11.1)	NS	48.7 (10.7)	28.5 (7.9)	0.003
ALT (U/l) (mean ± SD)	32.3 (9.3)	42.0 (11.5)	NS	37.7 (11.5)	26.8 (9.2)	0.044
AGA IgA (U/mL) (median; IQR)	265.0 (5.0–301.0)	154.4 (3.5–318.5)	NS	13.3 (6.0–14.4)	20.5 (3.5–21.5)	NS
Positive for AGA IgA (n, %)	5 (71.4)	4 (66.7)	NS	2 (28.6)	4 (66.7)	NS
AGA IgG (U/mL) (median; IQR)	231.0 (5.8–861.8)	154.4 (3.0–219.0)	NS	17.2 (3.5–113.0)	16.7 (1.5–19.3)	NS
Positive for AGA IgG (n, %)	6 (85.7)	5 (83.3)	NS	3 (42.9)	3 (50.0)	NS
tTg-IgA (U/mL) (median; IQR)	12,052.0 (3.0–13,565.0)	7013.7 (44.0–16,505.8)	NS	88.4 (3.3–414.3)	367.25 (2.0–503.0)	NS
Positive for tTg-IgA (n, %)	7 (100.0)	6 (100.0)	NS	5 (71.4)	5 (83.3)	NS
tTg-IgG (U/mL) (median; IQR)	1117.3 (6.25–1282.75)	97.5 (7.0–123.0)	NS	13.6 (4.5–32.5)	15.8 (3.0–22.0)	NS
Positive for tTg-IgG (n, %)	7 (100.0)	5 (83.3)	NS	3 (42.9)	3 (50.0)	NS
Positive for EMA (n, %)	7 (100.0)	5 (83.3)	NS	4 (57.1)	4 (66.7)	NS

AGA: anti-gliadin antibody; ALT: alanine aminotransferase; AST: aspartate aminotransferase; CD: celiac disease; EMA: anti-endomysial antibody; Ig: immunoglobulin; IQR: interquartile range; HT: hypertransaminasemia; NA: not applicable; NS: not significant; SD: standard deviation; tTg: tissue transglutaminase.

**Table 7 nutrients-17-00085-t007:** Studies analyzing correlations between CD, metabolic syndrome, and MAFLD in treatment-naive (i.e., gluten-containing diet) CD subjects.

Reference	Year	Study Design	Population	Results
Dickey et al. *Br. Med. J.* [105]	1998	Prospective study	50 adult CD patients	Prevalence of overweight: 28% Prevalence of obesity: 6%
Dickey et al. *Am. J. Gastroenterol*. [110]	2006	Retrospective study	371 CD patients	Prevalence of overweight: 26% Prevalence of obesity: 13%
Valletta et al. *Eur. J. Clin. Nutr.* [132]	2010	Retrospective study	149 pediatric CD patients	Prevalence of overweight: 11% Prevalence of obesity: 3%
Cheng J *J. Clin. Gastoenterol*. [133]	2010	Retrospective study	369 CD patients	Prevalence of overweight: 15.2% Prevalence of obesity: 6.8%
Reilly et al. *Pediatr. Gastroenterol. Nutr*. [109]	2011	Retrospective study	142 pediatric CD patients	Prevalence of overweight: 12.6% Prevalence of obesity: 6%
Tucker et al. *J. Gastroint. Liver Dis.* [107]	2012	Retrospective study	187 adult CD patients	Prevalence of overweight: 31% Prevalence of obesity: 13%
Ukkola et al. *Eur. J. Inter.* *Med.* [134]	2012	Prospective study	698 adult CD patients	Prevalence of overweight: 28% Prevalence of obesity: 11%
Norsa et al. *World J. Gastroenterol.* [135]	2013	Multicenter prospective study	114 child CD patients	Prevalence of obesity: 5.3% Prevalence of overweight: 8.8%
Tortora et al. *Aliment. Pharmacol. Ther.* [119]	2015	Observational prospective study	98 adult CD patients	Prevalence of MetS: 2% Prevalence of HS on US: 18%
Reilly et al. *J. Hepatol.* [14]	2015	Population-based cohort study	26,816 child and adult CD patients	Increased risk of NAFLD (OR 3.9, CI 95% 2.8–5.5) vs. 130,051 reference individuals
Singh et al. *Indian J. Gastroenterol*. [106]	2016	Retrospective study	210 adolescent and adult CD patients	Prevalence of overweight: 6.2% Prevalence of obesity: 2.9%
Stein et al. *J. Clin.* *Gastroenterol* [108]	2016	Retrospective study	258 CD patients	Prevalence of overweight/obesity: 38.3%
Ciccone et al. *Digestion* [129]	2019	Retrospective study	185 adult CD patients	Prevalence of MetS: 3.24% Prevalence of HS: 1.7%
Agarwal et al. *Intest. Res.* [128]	2021	Prospective study	44 adult CD patients	Prevalence of MetS: 11.4% Prevalence of HS on FibroScan^®^: 14.3% Prevalence of hyperglycemia: 11.9% Prevalence of hypertriglyceridemia: 9.1% Prevalence of overweight/obesity: 13.6% Prevalence of an elevated waist circumference: 22.7%
Rispo et al. *Liver Int.* [130]	2021	Retrospective study	221 CD adult patients	Prevalence of NAFLD: 29.4% Prevalence of MAFLD: 14.5%
Barone et al. *Nutr. Reviews* [136]	2023	Systematic review with a meta-analysis	7959 pediatric and adult CD patients vs. 20,524 healthy controls	Overall CD patients Prevalence of overweight: 14% Prevalence of obesity: 6% (RR 0.69; 95% CI, 0.57–0.83; *p* < 0.001) Prevalence of overweight: adults with CD 20% vs. pediatric patients with CD 9% (*p* < 0.001) Prevalence of obesity: adults with CD 10% vs. pediatric patients with CD 3% (*p* < 0.001)
Aggarwal et al. *Dig. Dis. Sci.* [131]	2024	Systematic review and meta-analysis	2578 adult CD patients	Prevalence of MetS: 4.3% (95% CI 2.4–6.7) Prevalence of HS: 18.2% (95% CI 8.3–30.8)

CD: celiac disease; CI: confidence interval; HS: hepatic steatosis; MAFLD: metabolic dysfunction-associated steatotic liver disease; MetS: metabolic syndrome; NAFLD: non-alcoholic fatty liver disease; OR: odds ratio; RR: relative risk.

**Table 8 nutrients-17-00085-t008:** Studies analyzing correlations between CD, metabolic syndrome, and MAFLD in subjects with CD after starting a GFD.

Reference	Year	Study Design	Population	Number of Years on a GFD	Results
Kabbani et al. Aliment. *Pharmacol. Ther.* [146]	2012	Retrospective study	679 adult CD patients	39.5 months (1–345)	BMI increased from 24.0 to 24.6
Valletta et al. *Eur. J. Clin. Nutr*. [132]	2012	Retrospective study	149 pediatric CD patients	4.5 years (1–16.3) (median and range)	The prevalence of overweight increased from 11% to 21% (*p* = 0.03)
Norsa et al. *World J. Gastroenterol*. [135]	2013	Multicenter, prospective study	114 pediatric CD patients	>1 year	Prevalence of 3 or more concomitant CVD risk factors: 14% Prevalence of hypertriglyceridemia: 34.8% Prevalence of hypertension: 29.4% Prevalence of high LDL levels: 24.1% Prevalence of insulin resistance: 3.5% The prevalence of borderline LDL cholesterol levels increased from 10% to 24% (*p* = 0.090) The prevalence of overweight increased from 8.8% to 11.5% The prevalence of obesity increased from 5.3% to 8.8%
Reilly et al. *J. Hepatol.* [14]	2015	Population-based cohort study	2816 pediatric and adult CD patients vs. 130,051 pediatric and adult controls		Increased risk of NAFLD
				1 year	HR 13.3 (95% CI 3.5–50.3, *p* < 0.001)
				5 years	HR 4.6 (95% CI 2.0–10.6, *p* < 0.001)
				>15 years	HR 2.5 (95% CI 1.0–5.9, *p* = 0.043)
Tortora et al. *Aliment. Pharmacol. Ther*. [119]	2015	Observational, prospective study	98 adult CD patients	1 year	The prevalence of MetS increased from 2% to 29.5% (OR 20, *p* < 0.01) The prevalence of HS increased from 18% to 28.5% (*p* = 0.1) Increased prevalence of an elevated waist circumference (OR 2.8, *p* < 0.01) Increased prevalence of high blood pressure (OR 5.2, *p* < 0.01) Increased prevalence of hyperglycemia (OR 4.4, *p* = 0.01) Increased prevalence of overweight (*p* = 0.01) Increased prevalence of hypertriglyceridemia (*p* = 0.05) Reduced prevalence of low HDL cholesterol levels (*p* = 0.7)
Tovoli et al. *Aliment. Pharmacol. Ther*. [15]	2018	Case–control study	202 CD patients versus 202 healthy controls	>6 months	Increased risk of NAFLD (OR 2.90, 95% CI 1.64–5.5, *p* < 0.001)
			Non-overweight CD patients versus non-overweight controls		Increased risk of NAFLD (OR 5.71, 95% CI 2.3–14.19, *p* < 0.001)
			Overweight/obese CD patients vs. overweight/obese controls		No difference in the prevalence of NAFLD
Ciccone et al. *Digestion* [129]	2019	Retrospective study	185 adult CD patients	7 (1–36) years (median and range)	The prevalence of MetS increased from 3.24% to 14.6% (*p* < 0.0001) The prevalence of HS increased from 1.7% to 11.1% (*p* < 0.0001)
Agarwal et al. *Intest. Res.* [128]	2021	Prospective study	44 adult CD patients	1 year	The prevalence of MetS increased from 11.4% to 18.2% (*p* = 0.219) The prevalence of HS on FibroScan^®^ increased from 14.3% to 29.5% (*p* = 0.002) The prevalence of hyperglycemia increased from 11.9% to 30.9% (*p* = 0.039) The prevalence of overweight/obesity increased from 13.6% to 25% (*p* = 0.003)
		Retrospective study	130 adult CD patients	4 years (median)	Prevalence of Mets: 26.3% Prevalence of HS: 23%
Rispo et al. *Liver Int.* [130]	2021	Retrospective study	221 CD patients	2 years	The prevalence of NAFLD increased from 29.4% to 46.6% The prevalence of MAFLD increased from 14.5% to 32.6%
Aggarwal et al. *Dig. Dis. Sci*. [131]	2024	Systematic review and meta-analysis	2578 CD patients	From 6 months to 36 years	The prevalence of MetS increased from 4.3% (95% CI 2.4–6.7) to 21.3% (95% CI 11.7- 32.9) The prevalence of HS increased from 18.2% (95% CI 8.3–30.8) to 28% (95% CI 20.7–36.4)

CD: celiac disease; CI: confidence interval; CVD: cardiovascular disease; GFD: gluten-free diet; HDL: high-density lipoprotein; HR: hazard ratio; HS: hepatic steatosis; LDL: low-density lipoprotein; MAFLD: metabolic dysfunction-associated steatotic liver disease; MetS: metabolic syndrome; NAFLD: non-alcoholic fatty liver disease; OR: odds ratio.

## Data Availability

The data presented in this study are available upon request from the corresponding author. The data are not publicly available due to privacy restrictions.

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
