# Peer review of "Celiac Disease and Liver Damage: The Gut–Liver Axis Strikes Back (Again)? A Retrospective Analysis in the Light of a Literature Review"

_nutrients, 2024, doi:10.3390/nu17010085_

Round 1

Reviewer 1 Report

Comments and Suggestions for Authors

After careful consideration, I fell that the manuscript entitled “Celiac disease and liver damage: the gut-liver axis strikes back (again)? A retrospective analysis in the light of a literature review” has merit, but is not suitable for publication as it currently stands.

Here are my comments:

1. Section 2.2.2:
- The exclusion criteria “Age <0.5 and >=15 years” is not necessary, as children with these characteristics do not satisfy the inclusion criteria.

2. Statistical Analysis section:
-  the Kruskall-Wallis test was not used in the study. So the description of this test should be removed;
- I suggest describing the test used to verify the normality of the data;
- Describe in this section the level of significance adopted in the statistical tests
- Describe the paired tests performed: paired t-test, Wilcoxon or Mc Nemar test (see Comment 6.)

3. Section 3.1:
- The Supplementary Table 1 was not cited in the manuscript. It can be cited in the first paragraph of this section to indicate the source of the results presented.
- the authors state that “Only 26 subjects were required to undergo EDS plus biopsy to confirm the diagnosis (or rule out the coexistence of other conditions), thus the 'no biopsy-approach' was applied in 83.8% of the children enrolled (see Supplementary Table 3)”. It is not clear where this information can be extracted from the Supplementary Table 3.

4. Section 3.2:
- It is not difficult to see in Table 1 when the authors say that: “Analysis of the baseline features of the patients with hypertrans-aminasemia compared to all the other CD children, showed a higher prevalence within the first year and a half of life (P=0.02)…” (second paragraph of this section). In fact, this reading would be easier if the percentages of the categorical variables were presented in rows (instead in columns). I suggest that the authors implement this change (for the other tables too).

5. Table 1:
- The tests (chi-square) for variables with more than three levels are incorrect. In fact, these tests cannot be performed separately for each level (row) of the variable. For example, the variable “Age at CD diagnosis” results in a 3x2 table with a single p-value from the chi-squared test. Therefore, the tests for the variables: age at CD diagnosis, BMI class, type of CD, stool changes, autoimmune disease and Marsh score (Table 2) should be revised.

6. Table 2:
- Typo in the variables MCH and RDH (“,” in decimals). Check all tables for typos.
- Indicate the IQR in the footnote, in the same way as SD. The same for Tables 3-5.

7. Table 4:
- The tests in Table 4 are incorrect. In fact, statistical tests should consider paired groups: paired t-test for normal data, Wilcoxon test for non-normal and the Mc Nemar test for dichotomous variables.

8. Supplementary Table 5:
- Indicate the sample size N;
- Check the calculated values, since some have small errors. For example, 25/160 = 15.6% (Positive AGA IgA). Check all the other tables too.

Author Response

Dear Editor,

We are happy that our paper has been considered as a potential publication on your prestigious Journal.

We would like to thank the reviewers for the valuable suggestions they gave us to improve our paper. We greatly appreciated their care in the revision, and we tried to modify the paper according to all their suggestions. We hope that, in this way we have improved the manuscript.

Here follows a point-to-point response to all the comments.

Best regards

Reviewer 1

After careful consideration, I fell that the manuscript entitled “Celiac disease and liver damage: the gut-liver axis strikes back (again)? A retrospective analysis in the light of a literature review” has merit, but is not suitable for publication as it currently stands.

R: we thank the reviewer for her/his valuable comments and for the deep effort he put into analyzing our paper. We fully agree with her/him that some changes are needed, and we have tried to modify point by point all the weaknesses she/he kindly highlighted.

Here are my comments:

1. Section 2.2.2:
- The exclusion criteria “Age <0.5 and >=15 years” is not necessary, as children with these characteristics do not satisfy the inclusion criteria.

R: we thank the reviewer, and we modified removing this exclusion criteria.

2. Statistical Analysis section:
-  the Kruskall-Wallis test was not used in the study. So the description of this test should be removed;
- I suggest describing the test used to verify the normality of the data;
- Describe in this section the level of significance adopted in the statistical tests
- Describe the paired tests performed: paired t-test, Wilcoxon or Mc Nemar test (see Comment 6.)

R: we thank the reviewer for her/his suggestions. We apologize for the errors made in the first version of the work. Most of the errors highlighted by the reviewer were due to an error related to the progressive analysis of the data. In fact, the more we proceed with the analysis, the more we changed the types of statistical analysis performed, but, unfortunately, at the time of submission we forgot to update this section, and we left the 'Version 1.0'. We apologize, again, for the error and we have modified point by point as follows.

- I suggest describing the test used to verify the normality of the data

R: we added: ‘Normality was assessed with the Shapiro–Wilk test’.

- Describe in this section the level of significance adopted in the statistical tests

R: we added: ‘Values of p<0.05 were statistically significant’.

- Describe the paired tests performed: paired t-test, Wilcoxon or Mc Nemar test (see Comment 6.)

R: as suggested in Comment 7, now we have applied the Mc Nemar test to recalculate significance for paired proportions. We modified the text as follows: ‘The paired tests were performed with: T Student for paired test when the distribution was Gaussian otherwise with Wilcoxon test.

The χ2 test and Fisher’s exact test were used to compare frequency values in the various population groups, McNemar test was used to compare paired proportions’.

3. Section 3.1:
- The Supplementary Table 1 was not cited in the manuscript. It can be cited in the first paragraph of this section to indicate the source of the results presented.

R: we thank the reviewer. We are sorry for this mistake and we modified the text adding the citation of Supplementary Table 1 as follows: ‘The majority of them presented with the typical manifestations of CD (50.6%), although only a minority (18.1%) reported diarrhea, but also extraintestinal manifestations (25.0%) and anemia (22.5%) were recorded (see Supplementary Table 1).’

- the authors state that “Only 26 subjects were required to undergo EDS plus biopsy to confirm the diagnosis (or rule out the coexistence of other conditions), thus the 'no biopsy-approach' was applied in 83.8% of the children enrolled (see Supplementary Table 3)”. It is not clear where this information can be extracted from the Supplementary Table 3.

R: we are sorry for this misunderstanding, and we thank the reviewer to allow us to clarify this point. We thought that by reporting the Marsh degrees of subjects who underwent EDS, it was possible to deduce how many subjects had undergone a biopsy and how many had not. Obviously, this is linked to an 'overconfidence' with our research, and what is clear to us may be, for sure, not clear to a reader. Thus, we have modified Supplementary Table 3.

4. Section 3.2:
- It is not difficult to see in Table 1 when the authors say that: “Analysis of the baseline features of the patients with hyper-transaminasemia compared to all the other CD children, showed a higher prevalence within the first year and a half of life (P=0.02)…” (second paragraph of this section). In fact, this reading would be easier if the percentages of the categorical variables were presented in rows (instead in columns). I suggest that the authors implement this change (for the other tables too).

R: we thank the reviewer for her/his suggestion. We strongly considered modifying the reporting of percentages for categorical variables in rows, but after recalculating and modifying the tables, we thought the reading could be misleading. Having reported the percentages in columns rather than rows, in our humble opinion, better matches the message we would like to give to potential readers of our research work. In fact, by reporting the percentages in columns we think it is easier to understand what is the prevalence of ‘presence/absence - positivity/negativity’ of a specific variable, compared to the number of subjects who present the specific 'outcome variable' that we are considering in the single table (e.g. 'Persistent HT' vs 'Non-Persistent HT' or 'No US Liver Steatosis' vs 'US Liver Steatosis’, etc.). By always keeping the denominator of the proportion the same (i.e. number of patients who present the specific 'outcome variable'), in fact, we hypothesize that the readers of our paper will be able to better focus on the message we intend to convey. In light of the above, thanking the reviewer again for her/his much-appreciated suggestion, which led us to reflect on a possible further 'vulnus' of our work, we would suggest to leave the reporting of percentages unchanged.

5. Table 1:
- The tests (chi-square) for variables with more than three levels are incorrect. In fact, these tests cannot be performed separately for each level (row) of the variable. For example, the variable “Age at CD diagnosis” results in a 3x2 table with a single p-value from the chi-squared test. Therefore, the tests for the variables: age at CD diagnosis, BMI class, type of CD, stool changes, autoimmune disease and Marsh score (Table 2) should be revised.

R: we thank the reviewer for this extremely important comment. We are sorry for the mistake, and we strongly agree with the reviewer thus we reanalyzed the variables which had 3 or more levels, reporting, now the ‘whole’ statistical significance for the analyzed variable. According to the new significance we have modified the text in several points (Abstract, Results, Discussion and Conclusion), removing the no-more significative data.

6. Table 2:
- Typo in the variables MCH and RDH (“,” in decimals). Check all tables for typos.

R: we are sorry for this oversight. We checked and modified through the tables and the entire text.
- Indicate the IQR in the footnote, in the same way as SD. The same for Tables 3-5.

R: we thank the reviewer for this comment; we modified as required.

7. Table 4:
- The tests in Table 4 are incorrect. In fact, statistical tests should consider paired groups: paired t-test for normal data, Wilcoxon test for non-normal and the Mc Nemar test for dichotomous variables.

R: we thank the reviewer for pointing out this error. In fact, if we had correctly used the paired t-test for normal data and the Wilcoxon test for non-normal data, but we had forgotten to apply the Mc Nemar test for the study of paired proportions. We have recalculated all the significances for the latter and added a new table (Table 4 bis) reporting the results of the Mc Nemar test.

8. Supplementary Table 5:
- Indicate the sample size N;

R: we thank the reviewer, and we added the sample size as required.

- Check the calculated values, since some have small errors. For example, 25/160 = 15.6% (Positive AGA IgA). Check all the other tables too.

R: we thank the reviewer; we checked all the tables and we modified all transcription errors.

Reviewer 2 Report

Comments and Suggestions for Authors

This is a retrospective analysis and literature review, investigating the prevalence of hypertransaminasemia and hepatic steatosis (HS) in a pediatric population with celiac disease (CD) before starting a gluten-free diet (GFD) to analyze how the introduction of GFD could modify this condition. The authors concluded that GFD can reverse the trend of liver involvement, although periods of longer than 12 months may be necessary. However, GFD has been associated with an increased prevalence of HS, especially in adult CD patients, so a regular follow-up involving a nutritionist should be recommended to guide physicians in patient management. In general, the analysis was well-performed and the manuscript was well-written.

Detailed comments:

1) Not all abbreviations have been introduced with full names at the first time of appearance, such as MASLD and anti-tTg IgG in Abstract, HBV and HCV in Page 3, AGA and EMA in Page 5, LMER in Page 12 and IEL in Figure 2.

2) What is the "Adherence (Biagi/Pavia) score" in Table 4? Adherence to what?

3) Why was Figure 2 placed in the middle of the manuscript? It is a schematic diagram. Pathogenic links between CD and metabolic dysfunction-associated steatotic liver disease (MASLD) development should work on both pediatric and adult patients. Thus, the schematic diagram should be placed at the end of the manuscript.

4) Also for Figure 2, besides cytokine storm, other pathogenic mechanisms should also be discussed.

Author Response

Dear Editor,

We are happy that our paper has been considered as a potential publication on your prestigious Journal.

We would like to thank the reviewers for the valuable suggestions they gave us to improve our paper. We greatly appreciated their care in the revision, and we tried to modify the paper according to all their suggestions. We hope that, in this way we have improved the manuscript.

Here follows a point-to-point response to all the comments.

Best regards

Reviewer 2

This is a retrospective analysis and literature review, investigating the prevalence of hypertransaminasemia and hepatic steatosis (HS) in a pediatric population with celiac disease (CD) before starting a gluten-free diet (GFD) to analyze how the introduction of GFD could modify this condition. The authors concluded that GFD can reverse the trend of liver involvement, although periods of longer than 12 months may be necessary. However, GFD has been associated with an increased prevalence of HS, especially in adult CD patients, so a regular follow-up involving a nutritionist should be recommended to guide physicians in patient management. In general, the analysis was well-performed and the manuscript was well-written.

R: we thank the reviewer for her/his appreciation. She/He fully understood the meaning of our research work and the message we would like to address to the readers if our paper will be considered suitable for publication.

Detailed comments:

  • Not all abbreviations have been introduced with full names at the first time of appearance, such as MASLD and anti-tTg IgG in Abstract, HBV and HCV in Page 3, AGA and EMA in Page 5, LMER in Page 12 and IEL in Figure 2.

R: we are sorry for these mistakes; we checked and modified throughout the text. For what concerns AGA and EMA, they were introduced in page 3. In addition, we removed LMER because it was not useful to use this abbreviation.

  • What is the "Adherence (Biagi/Pavia) score" in Table 4? Adherence to what?

R: we thank the reviewer for this comment and we modified both table 4 and supplementary table 4 using the term ‘Biagi/Pavia GFD adherence score’:

  • Why was Figure 2 placed in the middle of the manuscript? It is a schematic diagram. Pathogenic links between CD and metabolic dysfunction-associated steatotic liver disease (MASLD) development should work on both pediatric and adult patients. Thus, the schematic diagram should be placed at the end of the manuscript.

R: we thank the reviewer for this suggestion. We agree that figure 2 might be moved at the end of the literature revision. We introduced it as follows: ‘Figure 2 summarizes the possible mechanisms that might be involved in MASLD development in CD patients both before and after the introduction of the GFD’.

  • Also for Figure 2, besides cytokine storm, other pathogenic mechanisms should also be discussed.

R: we thank the reviewer for this comment. We have modified figure 2 adding other possible mechanisms responsible for MASLD development in CD both before and after adoption of GFD.

Reviewer 3 Report

Comments and Suggestions for Authors

This an interesting paper. For me the paper’s topić is New. In my opinion only introduction should be improved. Please describe research gap and hypothesis.

Author Response

Dear Editor,

We are happy that our paper has been considered as a potential publication on your prestigious Journal.

We would like to thank the reviewers for the valuable suggestions they gave us to improve our paper. We greatly appreciated their care in the revision, and we tried to modify the paper according to all their suggestions. We hope that, in this way we have improved the manuscript.

Here follows a point-to-point response to all the comments.

Best regards

Reviewer 3

This an interesting paper. For me the paper’s topić is New. In my opinion only introduction should be improved. Please describe research gap and hypothesis.

R: we thank the reviewer for her/his opinion on our research work. We agree with what she/he pointed out and we have modified the introduction trying to better explain what was requested as follows:

‘Despite the growing number of evidence, to date, few studies have focused on this condition in pediatric age, some reporting that the prevalence of CD among children with NAFLD could be comparable to the prevalence of CD in the general pediatric population [21]. Nevertheless, a study conducted on 11,488 children (0-19 years) affected by CD compared with 57,029 healthy children (of the same age and sex) supports the hypothesis of a close association between CD and NAFLD in childhood [14]. Even less evidence about possible pathogenic mechanisms and/or pre or post-GFD variables associated with MASLD development in CD children has been reported.

In this scenario, we wanted to analyze how this specific condition is characterized, both before and after GFD, in a CD pediatric population at-tending the main third level centers for the diagnosis and treatment of CD in western Sicily (Italy)’.

Reviewer 4 Report

Comments and Suggestions for Authors

This retrospective study analyzed the prevalence of hypertransaminasemia and hepatic steatosis (HS) in Sicilian pediatric patients with celiac disease (CD) before and after introducing a gluten-free diet (GFD). The findings revealed that while GFD reduced the prevalence of hypertransaminasemia and HS in some cases, a significant percentage of patients exhibited persistence of these conditions after 12 months. The study also performed a literature review to contextualize these observations in the broader framework of CD-related liver involvement.

This is an interesting and valuable study highlighting the gut-liver axis's role in CD-related liver damage. However, both the abstract and the full paper feel overly long and repetitive, diluting the impact of the findings. Summarizing the results and key discussions more concisely could enhance readability and clarity.

Did the authors evaluate amylase and lipase levels in the patients? Pancreatic enzyme elevations have been documented in some cases of CD and could provide additional insights into the broader spectrum of gastrointestinal involvement.

Additionally, was refractory celiac disease (RCD) and its subtypes (e.g., type I vs. type II) analyzed in the context of liver damage? These details would help understand whether persistent liver abnormalities might correlate with more severe or atypical forms of CD.

Author Response

Dear Editor,

We are happy that our paper has been considered as a potential publication on your prestigious Journal.

We would like to thank the reviewers for the valuable suggestions they gave us to improve our paper. We greatly appreciated their care in the revision, and we tried to modify the paper according to all their suggestions. We hope that, in this way we have improved the manuscript.

Here follows a point-to-point response to all the comments.

Best regards

Reviewer 4

This retrospective study analyzed the prevalence of hypertransaminasemia and hepatic steatosis (HS) in Sicilian pediatric patients with celiac disease (CD) before and after introducing a gluten-free diet (GFD). The findings revealed that while GFD reduced the prevalence of hypertransaminasemia and HS in some cases, a significant percentage of patients exhibited persistence of these conditions after 12 months. The study also performed a literature review to contextualize these observations in the broader framework of CD-related liver involvement.

R: we thank the reviewer for her/his analysis and for appreciating our paper, which tries, in fact, to contextualize the data of our population within a global panorama on the relationship between intestine - CD - liver - metabolic disease.

This is an interesting and valuable study highlighting the gut-liver axis's role in CD-related liver damage. However, both the abstract and the full paper feel overly long and repetitive, diluting the impact of the findings. Summarizing the results and key discussions more concisely could enhance readability and clarity.

R: we thank the reviewer again for her/his suggestions and we fully understand that the text may appear long. However, it is the very nature of our research work, which not only reports original data but associates them with an extensive review of the literature, that determines its length. In fact, since the topic is quite debated and with multiple studies in adults, and with controversial opinions in children, and having included both the pre and post-GFD data, we have tried to report as faithfully as possible what is reported in the literature. It is plausible that in this examination some concepts may be repeated in some points, but these are introductory to what is presented later and eliminating them would leave part of the text 'truncated'.

To avoid the possibility of information dispersion, we have created summary tables and have consistently modified figure 2, summarizing in a single diagram all the main connectors between CD, GFD, hepatic steatosis and MASLD. We have also removed some sentences from the abstract to streamline the text.

We hope that these changes will respond to the reviewer's requests and, for sure, improve our paper.

Did the authors evaluate amylase and lipase levels in the patients? Pancreatic enzyme elevations have been documented in some cases of CD and could provide additional insights into the broader spectrum of gastrointestinal involvement.

R: we thank the reviewer for this very interesting comment. Unfortunately, we are not able to provide data regarding pancreatic indices because these, although often requested in our patients, are not reported in the evaluation forms, except in cases of significant alterations. Specifically, we re-checked the forms of the patients included in our study and we highlighted that in only two patients, both without steatosis and without hypertransaminasemia, the values ​​of amylase and lipase were reported; therefore, no additional analysis could be conducted on this aspect.

Additionally, was refractory celiac disease (RCD) and its subtypes (e.g., type I vs. type II) analyzed in the context of liver damage? These details would help understand whether persistent liver abnormalities might correlate with more severe or atypical forms of CD.

R: we thank the reviewer for this comment. In our multicentric case series, we have never found children affected by refractory CD. The patients in our study who did not have an improvement in symptoms at follow-up were all patients who were not adequately adherent to the GFD. We understand, however, that the data could be interesting and, if not specified among the exclusion criteria, a reader could be misled into thinking that some patients may be affected by refractory CD, thus persisting with elevated transaminase’s levels. Therefore, we have added the following sentence to the exclusion criteria: ‘Type I and type II refractory CD diagnosis’.

Round 2

Reviewer 1 Report

Comments and Suggestions for Authors

I feel that the authors have corrected and/or justified the questions satisfactorily. I therefore believe that the work is now suitable for publication. I have no additional comments.